



**CO₂ flux over young and snow-covered Arctic sea ice in**
**winter and spring**
Daiki Nomura[1, 2, 3*], Mats A. Granskog[4], Agneta Fransson[4], Melissa Chierici[5, 6], Anna
Silyakova[7], Kay I. Ohshima[1, 3], Lana Cohen[4], Bruno Delille[8], Stephen R. Hudson[4], and
Gerhard S. Dieckmann[9]
1 Institute of Low Temperature Science, Hokkaido University, Kita–19, Nishi–8, Kita–
ku, Sapporo, Hokkaido 060–0819, Japan.
2 Faculty of Fisheries Sciences, Hokkaido University, 3–1–1, Minato–cho, Hakodate,
Hokkaido 041–8611, Japan.
3 Arctic Research Center, Hokkaido University, Kita–21, Nishi–11, Kita–ku, Sapporo,
Hokkaido 001–0021, Japan.
4 Norwegian Polar Institute, Fram Centre, NO–9296 Tromsø, Norway.
5 Institute of Marine Research, NO–9294, Tromsø, Norway.
6 FRAM-High North Research Centre for Climate and the Environment, Tromsø,
Norway.
7 CAGE, Centre for Arctic Gas Hydrate, Environment and Climate, Tromsø, Norway.
8 Unité d'Océanographie Chimique, Université de Liège, Liège, Belgium.
9 Alfred Wegener Institute for Polar and Marine Research, Bremerhaven, Germany.



＊Corresponding author: Daiki Nomura, e-mail: daiki.nomura@fish.hokudai.ac.jp,
Faculty of Fisheries Sciences, Hokkaido University, 3–1–1, Minato–cho, Hakodate,
Hokkaido 041–8611, Japan.





**Abstract**

We show that young, snow-covered ice has a potential for sea-ice-to-air $CO_2$ release
during winter and spring in the Arctic Ocean north of Svalbard. Young thin sea ice was
characterized by high salinities and thus porosity, while the surface of thicker sea ice
was relatively warm (>–7.5°C), due to a thick insulating snow cover, even though air
temperatures were as low as –40°C. During these conditions, brine volume fractions of
sea ice were high, providing potentially favorable conditions for gas exchange between
sea ice and overlying air even in mid-winter. Although the potential $CO_2$ flux through
the sea ice decreased due to the presence of the snow, the snow surface still is a $CO_2$
source to the atmosphere for low snow density and thin snow conditions. Especially
young ice formed in leads, without snow cover, is important for the $CO_2$ flux from the
ice pack as the fluxes are an order of magnitude higher than for snow-covered older ice.



**1    Introduction**

Arctic sea ice is changing dramatically, with rapid declines in summer sea ice extent
and a shift towards younger and thinner first-year ice rather than thick multi-year ice
(e.g., Stroeve et al., 2012; Meier et al., 2014; Lindsay and Schweiger, 2015). Although
the effects of sea ice formation and melting on biogeochemical cycles in the ocean have
previously been discussed (e.g., Vancoppenolle et al., 2013), the effects of sea ice
freezing and melting on the carbon dioxide ($CO_2$) exchange with the atmosphere are
still large unknowns (Parmentier et al., 2013). Recent $CO_2$ flux measurements on sea ice
indicate that sea ice is an active component in gas exchange between ocean and
atmosphere (Nomura et al., 2013; Geilfus et al., 2013; 2014; Delille et al., 2014; Brown
et al., 2015; Kotovitch et al., 2016). However, due to the difficulty in acquiring
observations during winter, there is a definite lack of information on conditions during
wintertime.



The sea ice $CO_2$ fluxes depend on (a) the difference in the partial pressure of $CO_2$
($pCO_2$) between the sea ice surface and air, (b) ice surface condition including the snow
deposited on ice, and (c) wind-driven pressure pumping through the snow. For (a), it is
known that the air–sea ice $CO_2$ flux is driven by the differences in $pCO_2$ between the
sea ice surface and atmosphere (e.g. Delille et al., 2014; Geilfus et al., 2014). The brine
$pCO_2$ changes due to processes within the sea ice, such as thermodynamic process (e.g.,
Delille et al., 2014), biological activity (e.g., Delille et al., 2007; Fransson et al., 2013;
Rysgaard et al., 2013), and calcium carbonate ($CaCO_3$; ikaite) formation and dissolution
(e.g., Papadimitriou et al., 2012). When the $pCO_2$ in the brine is higher than that of the
air $pCO_2$, brine has the potential to release $CO_2$ to the atmosphere. For (b), the air–sea
ice $CO_2$ flux is strongly dependent on the sea ice surface conditions (Nomura et al.,
2010a, 2013; Geilfus et al., 2013; 2014; Barber et al., 2014; Brown et al., 2015;
Fransson et al., 2015). Nomura et al. (2013) proposed that snow conditions (e.g., water
equivalent) are important factors affecting gas exchange processes on sea ice. For (c), it
is thought that for snow cover, the $CO_2$ flux is affected by wind pumping (Takagi et al.,
2005) in which the magnitude of $CO_2$ flux through snow or overlying soil (e.g., Takagi
et al., 2005) increases due to wind pumping and can increase the transport with
molecular diffusion by up to 40% (Bowling and Massman, 2011). These results were
mainly found over land-based snow (soil and forest), and thus these processes are not
well understood over sea ice (Papakyriakou and Miller, 2011).

In addition to the processes described above, the $CO_2$ flux over sea ice may also be
influenced by the temperature difference between the ice surface and the atmosphere.
This has been shown in previous studies in dry snowpacks over land surfaces. These
studies show that there is an unstable air density gradient due to heating at the bottom
producing a strong temperature difference between bottom and top of snow (e.g.,
Powers et al., 1985; Severinghaus et al., 2010). This produces air flow within the
snowpack, which is a potentially significant contributor to mixing and transport of gas
and heat within the snowpack. We expect that this process would also occur in snow
over sea ice, especially during the wintertime when air temperatures are coldest and the
temperature difference between sea ice surface (snow bottom) and atmosphere is largest
(e.g., Massom et al., 2001). Generally, the sea ice surface under thick snow cover is



warm due to the heat conduction from the bottom of sea ice and the insulation effect of
the snow cover, and a strong temperature difference between sea ice surface and
atmosphere is observed (e.g., Massom et al., 2001). Such a temperature difference
would produce an unstable air density gradient and upward transport of air containing
$CO_2$ degassed at the sea-ice surface, thereby enhancing $CO_2$ exchange between sea ice
and atmosphere.

The air–sea ice $CO_2$ flux was examined using flux chambers on Arctic pack ice north of
Svalbard from mid-winter to spring (January to May 2015) during the Norwegian young
sea ICE (N-ICE2015) campaign to understand the air–sea ice $CO_2$ flux during cold
season, and effects of snow-cover on the air–sea ice $CO_2$ flux.



**2    Materials and Methods**

**2.1    Study site**

This study was performed during N-ICE2015 campaign with R/V Lance in the pack ice
north of Svalbard from January to June 2015 (Granskog et al., 2016). Air–sea ice $CO_2$
flux measurements were carried out from January to May 2015 during the drift of Floes
1, 2, and 3 of the N-ICE2015 campaign (Figures 1 and 2, Table 1). The ice pack was a
mixture of first-year ice and second-year ice (Granskog et al., 2017) both with a thick
snow cover (Merkouriadi et al., 2017). In the N-ICE2015 study region modal ice
thickness was about 1.3–1.5 m and modal snow thickness was almost 0.5 m (Rösel et al.,
2016a and b). Formation of leads and their rapid refreezing provided us the opportunity
to examine air–sea ice $CO_2$ fluxes over thin sea ice, occasionally covered with frost
flowers (Figure 2 and Table 1). Air temperature and wind speed were measured at a 10
m weather mast on the ice floe installed about 400 m away from R/V Lance (Cohen et
al., 2017).




**2.2    CO$_2$ flux measurements**

The air–sea ice CO$_2$ flux was measured with LI-COR 8100-104 chambers connected to
the LI-8100A soil CO$_2$ flux system (LI-COR Inc., USA) (Figure 2). This enclosed
chamber method has been widely applied over snow and sea ice (e.g., Schindlbacher et
al., 2007, Geilfus et al., 2015). Two chambers were connected in a closed loop to the
infrared gas analyzer (LI-8100A, LI-COR Inc., USA) to measure CO$_2$ concentration
through the multiplexer (LI-8150, LI-COR Inc., USA) with an air pump rate at 3 L min$^{-1}$
. Electricity was supplied by a car battery (8012-254, Optima Batteries Inc., USA).
Four CO$_2$ standards (324–406 ppmv) traceable to the WMO scale (Inoue and Ishii,
2005) were prepared to calibrate the CO$_2$ gas analyzer prior to the observations. CO$_2$
flux was measured from morning or afternoon during low-wind conditions (Table 2), to
minimize the effect of wind on the flux.

One chamber was installed over undisturbed snow or frost flowers over the ice surface.
The chamber collar was inserted 5 cm into the snow and 1 cm into ice at frost flowers
site to avoid air leaks between inside and outside of chamber. The second chamber was
installed after removing the snow or frost flowers. Flux measurements was begun
immediately in order to minimize the changes of the ice surface condition, and the data
of first CO$_2$ flux measurement was used. In order to evaluate the effect of removing
snow on sea ice surface temperature, ice surface temperature was monitored during CO$_2$
flux measurements at station FI6. To measure the sea ice surface temperature,
temperature sensor (RTR 52, T & D Corp., Japan) was installed in the top of the ice (1
cm) surface after snow removal. During first CO$_2$ flux measurements (about 30
minutes), ice surface temperature was stable at –5.8°C, suggesting that the effect of
removing snow on the variation of sea ice surface temperature was negligible. The
chamber was closed for 20 minutes in a sequence. The 20-minute time period was used
because CO$_2$ fluxes over sea ice are much smaller than over land. The CO$_2$
concentrations within the chamber were monitored to ensure that they changed linearly
throughout the measurement period. The CO$_2$ flux (mmol C m$^{-2}$ day$^{-1}$) (positive value
indicates CO$_2$ being released from air to ice surface) was calculated based on the
changes of the CO$_2$ concentration within the headspace of the chamber with LI-COR





software (Model: LI8100PC Client v.3.0.1.). The mean coefficient of variation for $CO_2$
flux measurements was less than 3.0% for $CO_2$ flux values larger than ±0.1 mmol C m$^{-2}$
day$^{-1}$. For $CO_2$ flux values smaller than ±0.1 mmol C m$^{-2}$ day$^{-1}$, the mean coefficient of
variation for $CO_2$ flux measurements was higher than 3.0%, suggesting that the
detection limit of this system is about 0.1 mmol C m$^{-2}$ day$^{-1}$.

In this paper, we express the $CO_2$ flux measured over the snow and frost flower as $F_{snow}$
and $F_{ff}$, respectively, and the flux measured directly over the sea ice surface either on
snow-free ice or after removal of snow and frost flower as $F_{ice}$. $F_{snow}$ and $F_{ff}$ are the
natural flux (snow and frost flowers are part of the natural system), and $F_{ice}$ is the
potential flux in cases when snow or frost flowers are removed. While removal of snow
and frost flowers is an artificial situation, comparisons between $F_{ice}$ and $F_{snow}$ or $F_{ff}$
provide information about the effect of snow on the $CO_2$ flux. Therefore, in this study,
we discuss both situations for $CO_2$ flux.


**2.3**     **Sampling of snow, frost flowers, brine, and sea ice**

For salinity measurements, snow was sampled, while frost flowers and surface of sea
ice after removing snow were sampled in bulk using a plastic shovel and taken into
plastic bag and placed in an insulated box for transport to the ship-lab for further
processing. These samples were melted slowly (2–3 days) in dark at +4°C. Temperature
of the snow and frost flowers were measured using a needle-type temperature sensor
(Testo 110 NTC, Brandt Instruments, Inc., USA). Accuracy of this sensor was ±0.2°C.
Snow density was obtained by a fixed volume sampler (Climate Engineering, Japan)
and weight measurement. The depth of the snow pack and frost flowers was also
recorded using a ruler.

Brine was collected for determination of salinity, dissolved inorganic carbon (DIC) and
total alkalinity (TA) for stations FI3–6 using the sackhole (Gleitz et al., 1995). First,
sackholes were drilled using an ice corer (Mark II coring system, Kovacs Enterprises,
Inc., USA) for 30 cm deep. The sackholes were covered with a lid made by 5 cm-thick



urethane to reduce heat and gas transfer between brine and atmosphere. When the brine
accumulated at the bottom of the sackholes (approximately 15 minutes), the brine was
collected with a plastic syringe (AS ONE Corporation, Japan) and kept in 500 mL
unbreakable plastic bottles (I-Boy, AS ONE Corporation, Japan) due to cold and harsh
conditions, as well as challenging transportation to the sampling sites. The brine bottle
without head-space was immediately put into an insulated box to prevent it from
freezing. Immediately after return to the ship, the brine samples were transferred to 250
mL borosilicate bottles (DURAN Group GmbH, Germany) for DIC and TA
measurements using tubing to prevent contact with air. The samples were preserved
with saturated mercuric chloride (HgCl$_2$, 60 μL for a 250 mL sample) and stored in dark
at +10°C until analyses at the Institute of Marine Research, Norway.

Sea ice was collected by same ice corer as described for brine collection at the same
location as snow and frost flowers were collected. Ice temperature was measured by
same sensor as described for snow on ice. Temperature sensor was inserted in holes
drilled into the core. Then, the core was cut with a stainless steel saw into 10 cm
sections for salinity and the ice sections placed into plastic bags. Sections were then
kept at +4°C and melted in the dark.


**2.4    Sample analysis**

Salinities for melted-snow, -frost flowers, -sea ice, and brine were measured with a
conductivity sensor (Cond 315i, WTW GmbH, Germany). For calibration of salinity
measurement, a salinometer Guildline PORTASAL Model 8410A, standardized by
International Association for the Physical Sciences of the Oceans (IAPSO) standard
seawater (Ocean Scientific International Ltd, UK) were used. Accuracy of this sensor
was ±0.003.

Analytical methods for DIC and TA determination are fully described in Dickson et al.
(2007). DIC in brine was determined using gas extraction of acidified sample followed
by coulometric titration and photometric detection using a Versatile Instrument for the

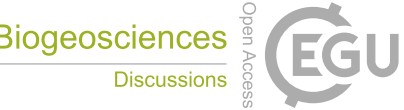

Determination of Titration carbonate (VINDTA 3C, Germany). TA of brine was
determined by potentiometric titration of 40 mL sample in open cell with 0.05 N
hydrochloric acid using a Titrino system (Metrohm, Switzerland). The average standard
deviation for DIC and TA, determined from replicate sample analyses from one sample,
was within $\pm 2$ µmol kg$^{-1}$ for both DIC and TA. Accuracy of the DIC and TA
measurements were $\pm 2$ µmol kg$^{-1}$ for both DIC and TA estimated using Certified
Reference Materials (CRM, provided by A. G. Dickson, Scripps Institution of
Oceanography, USA). The pCO$_2$ of brine (pCO$_{2\,b}$) was derived from in situ temperature,
salinity, DIC and TA of brine using the carbonate speciation program CO2SYS (Pierrot
et al., 2006). We used the carbonate dissociation constants ($K_1$ and $K_2$) of Mehrbach et
al. (1973) as refit by Dickson and Millero (1987), and the KSO$_4$ determined by Dickson
(1990). The conditional stability constants used to derived pCO$_2$ are strictly only valid
for temperatures above 0 °C and salinities between 5 and 50. Studies in spring ice
indicated that seawater thermodynamic relationships may be acceptable in warm and
low-salinity sea ice (Delille et al., 2007). In sea ice brines at even moderate brine
salinities of 80, Brown et al. (2014) found that measured and calculated values of the
CO$_2$ system parameters can differ by as much as 40%. On the other hand, because the
CO$_2$ system parameters are much more variable in sea ice than in seawater, sea ice
measurements demand less precision than those in seawater. Fransson et al. (2015)
performed one of few detailed analyses of the internal consistency using four sets of
dissociation constants and found that the deviation between measured and calculated
DIC varied between $\pm 6$ and $\pm 11$ µmol kg$^{-1}$, respectively. This error in calculated DIC
was considered insignificant in relation to the natural variability in sea ice.

The water equivalent was computed for snow by multiplying snow thickness by snow
density (Jonas et al., 2009). Brine volume of sea ice was calculated from the
temperature and salinity of sea ice according to Cox and Weeks (1983).



**3   Results**



### 3.1 Air temperature

Air temperature is shown in Figure 3. During the study period, air temperature varied
significantly from a low of –41.3°C (30 January) to a high of +1.7°C (15 June) (Hudson
et al., 2015). Even in wintertime (from January to March), rapid increases of air
temperature from below –30°C up to –0.2°C (e.g., 18 February), were observed. In
springtime (from April to June), the air temperature increased continuously, and from 1
June, air temperatures were near-constant 0°C, although rapid increases (and subsequent
decreases) of air temperature to near 0°C were observed on two occasions in mid-May
(Cohen et al., 2017).

### 3.2 Characteristics of snow, sea ice, and frost flower

The snow and ice thickness at the observation sites ranged between 0.0 and 60.0 cm and
between 15.0 and >200 cm, respectively (Table 1). The thin snow and ice represent
newly formed ice in leads. The thickness of the frost flowers ranged from 1.0 to 2.5 cm.

Figure 4 shows vertical profiles of snow and ice temperature and salinity in the top 20
cm of ice. Temperatures within the snowpack depended on the air temperature at the
time of observation. However, the bottom of the snow and the surface of the sea ice
were relatively warm (T>–7.5°C), except for the frost flower station YI1 and the multi-
year ice station OI1 (Figure 4a and Table 2). High salinities (S>18.6) characterized the
bottom of the snow and the surface of the sea ice, except for the multi-year ice station
OI1 (Figure 4b). At the multi-year ice station OI1, salinity was zero through the snow
and top of sea ice. Salinity of frost flowers was up to 92.8 for the thin ice station YI1
(Figure 4b). Snow density and water equivalent ranged from 268 to 400 kg m$^{-3}$ and 11
to 180 kg m$^{-2}$, respectively.

### 3.3 Physical and chemical properties of brine



The brine volume fraction, temperature, salinity, DIC, TA, and calculated $pCO_2$ are
summarized in Table 2. Brine volume fraction in top 20 cm of ice was from 9 to 17%,
except for the value of 0% at the multi-year ice station OI1 (Table 2). Brine
temperatures and salinity ranged from −5.3 to −3.3°C and 51.8 to 86.6, respectively.
DIC and TA of brine ranged from 3261 to 4841 µmol kg$^{-1}$ and 3518 to 5539 µmol kg$^{-1}$,
respectively. The $pCO_2$ of brine ($pCO_{2\,b}$) (334−693 µatm) was generally higher than
that of atmosphere ($pCO_{2\,a}$) (401 ± 7 µatm), except for station FI4.


**3.4    $CO_2$ flux**

Table 3 summarizes the $CO_2$ flux measurements for each surface condition. For
undisturbed natural surface conditions, i.e. measurements directly on the snow surface
($F_{snow}$) or the frost flowers ($F_{ff}$) on young ice, the mean $CO_2$ flux was +0.2 ± 0.2 mmol
C m$^{-2}$ day$^{-1}$ for $F_{snow}$ and +1.0 ± 0.6 mmol C m$^{-2}$ day$^{-1}$ for $F_{ff}$. The potential flux in
cases when snow or frost flowers had been removed ($F_{ice}$) was +2.5 ± 4.3 mmol C m$^{-2}$
day$^{-1}$. The air–sea ice $CO_2$ fluxes measured over the ice surface ($F_{ice}$) increased with
increasing difference in $pCO_2$ between brine and atmosphere ($\Delta pCO_{2\,b–a}$) with
significant correlation ($R^2 = 0.9$, $p < 0.02$), but this was not the case for $F_{snow}$ ($R^2 = 0.0$,
$p < 0.96$).



**4    Discussion**

**4.1    Effect of snow cover on the physical properties of sea ice surface**

In this study, we examined $CO_2$ fluxes between sea ice and atmosphere in a variety of
air temperature conditions from –32 to –3°C and diverse snow and ice conditions (Table
2). The bottom of the snow pack and the surface of the sea ice remained relatively warm
(>–7.5°C) (Figure 4a, Table 2) except for station OI1, even though air temperature was
sometimes below –40°C (Figure 3). Relatively warm ice temperatures were likely due



to the upward heat transport from the bottom of the ice and in cases the thick insulating
snow cover (Table 2). Therefore, snow acted as thermal insulator over sea ice, and in
general the snow depths observed during N-ICE2015 pointed towards this being
representative for first-year and second-year or older ice in the study region in winter
2015 (Rösel et al., 2016a). The young and first-year ice surfaces were characterized by
high salinities (Figure 4b). During sea ice formation, upward brine transport to the snow
pack occurs (e.g., Toyota et al., 2011). In addition, brine within the sea ice was not
completely drained as compared to that of multi-year ice. Furthermore, formation of
frost flowers and subsequent wicking up of surface brine into the frost flowers also
provides high salinity at the surface of sea ice (Kaleschke et al., 2004; Geilfus et al.,
2013; Barber et al., 2014; Fransson et al., 2015) as observed in this study (S>92)
(Figure 4b). Snowfall over the frost flowers would have preserved the high salinity at
the bottom of snow pack and top of sea ice for young and first-year ice.

As a result of the combination of the relatively high temperature and high salinity at the
top of sea ice, brine volume fractions in the upper parts of the sea ice were high, up to
17% (Table 2). It has been shown that ice permeability increases by an order of
magnitude when brine volume fraction > 5%, which would correspond to a temperature
of $-5°C$ for a bulk ice salinity of 5 – the so called "law of fives" (Golden et al., 1998;
Pringle et al., 2009; Zhou et al., 2013). Because sea ice temperatures was low and
thereby reduced permeability in winter season, generally, air–sea ice $CO_2$ flux is at its
minimum in the winter (e.g., Delille et al., 2014). However, in our study, the brine
volume fractions were generally >9%, except for station OI1 with fresh ice at the
surface, providing conditions for active gas exchange within sea ice and between sea ice
and atmosphere. This situation was likely made possible due to the thick snow cover
and relatively thin and young sea ice.


**4.2   $CO_2$ fluxes over different sea-ice surface types**

The $CO_2$ flux measurements over different surface conditions indicate that the snow on
sea ice affect the magnitude of air–sea ice $CO_2$ flux (Table 3). For undisturbed natural



surface conditions, the mean $CO_2$ flux measured directly over snow-covered first-year
ice and young ice with frost flowers ($F_{snow}$ and $F_{ff}$) was lower than the potential flux
obtained directly over the ice surface after removing snow ($F_{ice}$).

$F_{ff}$ indicates that the frost flower surface on young thin ice is a $CO_2$ source to the
atmosphere. Frost flowers are known to promote gas flux, such as $CO_2$, from the sea ice
to the atmosphere (Geilfus et al., 2013; Barber et al., 2014; Fransson et al., 2015). At
multi-year ice station OI1, neither snow or ice surface acted as a $CO_2$ source/sink. The
surface of multi-year ice did not contain any brine (Figure 4 and Table 2), and the top of
the ice was clear, colorless and very hard, suggesting superimposed formation at the top
of sea ice. This situation would be similar as for freshwater-ice and superimposed-ice as
these non-porous media block gas exchange effectively at the sea ice surface (Delille et
al., 2014). Snow-ice and superimposed-ice were frequently found in second-year ice
cores during N-ICE2015 (Granskog et al., 2017), so the 'blocking' of gas exchange in
second-year and multi-year ice may be a widespread process in the Arctic.

The magnitude of $F_{snow}$ is less than $F_{ice}$ (Table 3) indicating that the potential $CO_2$ flux
through sea ice decreased due to the presence of snow. Previous studies have shown
that snow accumulation over sea ice effectively impede $CO_2$ exchange (Nomura et al.,
2013; Brown et al., 2015). Nomura et al. (2013) reported that water equivalent of the
snow is an important factor that controls the $CO_2$ exchange through a snowpack.
Comparisons between stations FI5 and FI6 for $F_{snow}/F_{ice}$ ratio (0.2 for FI5 and 0.0 for
FI6) and water equivalent (11 kg m$^{-2}$ for FI5 and 127 kg m$^{-2}$ for FI6) indicate that the
$CO_2$ flux is affected by snow properties (density and depth). Although the potential $CO_2$
flux through the sea ice surface decreased by the presence of snow (Table 3), the snow
surface still presents a $CO_2$ source to the atmosphere for low snow density and shallow
depth conditions (e.g., +0.6 mmol C m$^{-2}$ day$^{-1}$ for FI5).


**4.3    Comparison to earlier studies on sea-ice to air $CO_2$ flux**





The $CO_2$ fluxes measured over the undisturbed natural surface conditions ($F_{snow}$ and $F_{ff}$)
in this study ranged from +0.1 to +1.6 mmol C $m^{-2}$ $day^{-1}$ (Table 3), which are at the
lower end of the reported range based on the chamber method and eddy covariance
method for natural and artificial sea ice (–259.2 to +74.3 mmol C $m^{-2}$ $day^{-1}$)
(Zemmelink et al., 2006; Nomura et al., 2006, 2010a, 2010b, 2013; Miller et al., 2011;
Papakyriakou and Miller, 2011; Geilfus et al., 2012, 2013; 2014; Barber et al., 2014;
Delille et al., 2014; Sørensen et al., 2014; Brown et al., 2015; Kotovitch et al., 2016).
Direct comparison to previous studies is complicated because $CO_2$ flux measurements
with both chamber and eddy covariance techniques were used during different condition
for season and ice surface characteristics. The footprint size of $CO_2$ exchange measured
with the two approaches (Zemmelink et al., 2006, 2008; Burba et al., 2008; Amiro,
2010; Miller et al., 2011; Papakyriakou and Miller, 2011; Sørensen et al., 2014; Miller
et al., 2015) may be one reason for the large difference.

When we compare our natural $CO_2$ flux range (+0.1 to +1.6 mmol C $m^{-2}$ $day^{-1}$ for $F_{snow}$
and $F_{ff}$) (Table 3) to estimates made by the chamber method in previous studies (–5.2 to
+6.7 mmol C $m^{-2}$ $day^{-1}$) (Nomura et al., 2006, 2010a, 2010b, 2013; Geilfus et al., 2012,
2013; 2014; Barber et al., 2014; Delille et al., 2014; Brown et al., 2015; Kotovitch et al.,
2016) (these studies include both natural and potential fluxes), our $CO_2$ fluxes are at the
lower end. However, our potential $CO_2$ flux ($F_{ice}$) was a larger $CO_2$ source (+11.8 mmol
C $m^{-2}$ $day^{-1}$) than reported in previous studies (+6.7 mmol C $m^{-2}$ $day^{-1}$). In our study,
the maximum potential flux (e.g., +11.8 mmol C $m^{-2}$ $day^{-1}$) was obtained for $F_{ice}$ at
station FI6 (Table 3). In this situation, $\Delta pCO_{2\,b-a}$ (293 µatm) was the highest (Table 2),
and it is reasonable to consider this as the highest magnitude of positive $CO_2$ flux within
our study. However, a previous study by closed chamber method showed that even for a
similar $\Delta pCO_{2\,b-a}$ (297 µatm) and magnitude for the brine volume fraction (10–15%),
the $CO_2$ flux was +0.7 mmol C $m^{-2}$ $day^{-1}$ for artificial sea ice with no snow in the tank
experiment (Nomura et al., 2006). In the following, we will discuss this difference.

The $CO_2$ flux (F) between the sea ice and overlying air can be expressed by the
following equation,



$F = r_b \, k \, \alpha \, \Delta pCO_{2 \, b-a}$,

where $r_b$ is the ratio of surface of the brine channel to sea ice surface, and we assume
that the value of $r_b$ is equal to brine volume fraction, k is the gas transfer velocity, $\alpha$ is
the solubility of $CO_2$ (Weiss, 1974), and $\Delta pCO_{2 \, b-a}$ is the difference in $pCO_2$ between
brine and atmosphere. The equation is based on the fact that $CO_2$ transfer between
seawater and air is controlled by processes in the near-surface water (Liss, 1973). The
gas transfer velocity (k) calculated from F, $r_b$, $\alpha$ and $\Delta pCO_{2 \, b-a}$ was 5.12 m day$^{-1}$ for $F_{ice}$
at station FI6 and 0.29 m day$^{-1}$ for the tank experiment examined in Nomura et al.
(2006). This result clearly indicates that the gas transfer velocity for $F_{ice}$ at station FI6 is
higher than that of tank experiment examined in Nomura et al. (2006) even with very
similar $\Delta pCO_{2 \, b-a}$ and brine volume fraction.

Here, we surmise that the gas transfer velocity and thereby $CO_2$ flux is greatly enhanced
by the temperature difference between sea ice surface and atmosphere. Previous studies
indicate that there is an unstable air density gradient in a dry snowpack due to basal
heating and the strong temperature difference develops between bottom and top of snow
(e.g., Powers et al., 1985; Severinghaus et al., 2010), which enhances the flow of air
through the snowpack. We propose that the mixing and transport of gas within the
snowpack could also occur over sea ice. Because temperatures at the bottom of snow
and the top of sea ice were relatively warm due to a thick insulating snow over sea ice,
there was a strong temperature difference between sea ice surface and atmosphere when
air temperature was low (Figure 4a and Table 2). For station FI6, temperature difference
between sea ice surface and atmosphere was 20.2°C after snow removal. On the other
hand, in the tank experiment by Nomura et al. (2006), the temperature difference
between sea ice surface (top 1.5 cm) and air in the headspace was only 4.5°C. Figure 5
shows the relationship between mean air–sea ice $CO_2$ fluxes and temperature difference
between ice and atmosphere. The strong dependence of $CO_2$ flux with temperature
difference ($T_{ice}-T_a$) was observed, especially for $F_{ff}$ and $F_{ice}$ ($R^2 > 0.7$, $p < 0.01$) (Figure
5). Due to the high brine volume fractions (Table 2), sea ice surface had enough
permeability for gas exchange. In addition, ice temperatures were similar for young and
first-year ice (Figure 5, Table 2), indicating that $pCO_2$ at the top of sea ice and $CO_2$ flux





would be of similar order of magnitude if thermodynamic processes dominated.
Therefore, our result suggest that the $CO_2$ fluxes even over the frost flower as natural
condition, would be enhanced by the upward transport of air containing high $CO_2$ from
the surface of sea ice to the atmosphere due to the strong temperature difference
between sea ice surface and atmosphere. Although the presence of snow on sea ice has
potential to produce a larger temperature difference between sea ice surface and
atmosphere and promote the upward transport, the magnitude of the $CO_2$ flux decreased
due to the presence of snow. However, for young sea ice likely the frost flower
conditions, ice surface temperature was warm (Table 2), suggesting that $CO_2$ flux would
be enhanced by the large temperature difference between sea ice surface and
atmosphere.



**5   Conclusions**

We measured $CO_2$ fluxes along with sea ice and snow physical and chemical properties
over first-year and young sea ice north of Svalbard in the Arctic ice pack. Our results
suggest that young thin snow-free ice, with or without frost flowers, is a source of
atmospheric $CO_2$ due to the high $pCO_2$ and salinity and relatively high sea ice
temperature. Although the potential $CO_2$ flux through the sea-ice surface decreased due
to the presence of snow, snow surface still presents a modest $CO_2$ source to the
atmosphere for low snow density and shallow depth situations. The highest ice to air
fluxes were observed over thin young sea ice formed in leads. During N-ICE2015 the
ice pack was dynamic, and formation of open water was associated with storms, where
new ice was formed. Open leads and storm periods were important for air-to-sea $CO_2$
fluxes (Fransson et al., 2017), due to undersaturation of the surface waters, while the
subsequent ice growth in these leads becomes important for the ice-to-air $CO_2$ fluxes in
winter due to the fact that the flux from young ice is an order of magnitude larger than
from snow-covered first-year ice.



High salinity and high sea ice temperature resulted in high brine volume fractions.
Given the fact that Arctic sea ice is shifting from multi-year ice to first-year ice (e.g.,
Stroeve et al., 2012; Meier et al., 2014), the area of thinner seasonal ice has increased.
Therefore, the amount of $CO_2$ released from sea ice surface to the atmosphere will be an
important fraction of the total $CO_2$ released by the Arctic Ocean. The dynamics of the
thinner ice pack, through formation of leads and new ice, will become an important in
the gas fluxes from the ice pack.



**6    Data availability**

Data used in this paper will be available at Norwegian Polar Data Centre
(data.npolar.no).



**7    Acknowledgments**

We would like to express heartfelt thanks to the crew of R/V Lance and all members of
the N-ICE2015 expedition for their support in conducting the field work. This work was
supported by the Japan Society for the Promotion of Science (#15K16135, #24-4175),
Research Council of Norway (KLIMAFORSK programme, grant 240639), the Centre
of Ice, Climate and Ecosystems (ICE) at the Norwegian Polar Institute through the N-
ICE project, the Ministry of Climate and Environment and the Ministry of Foreign
Affairs of Norway and the Grant for Joint Research Program of the Institute of Low
Temperature Science, Hokkaido University. AF, MC and MAG were supported by the
flagship research program "Ocean acidification and ecosystem effects in Northern
waters" within the FRAM-High North Research Centre for Climate and the
Environment.




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

Methods for Biogeochemical Studies of Sea Ice: The State of the Art, Caveats, and
Recommendation, Elementa, 3, 000038, doi:10.12952/journal.elementa.000038, 2015.

Nomura, D., Inoue, H. Y., and Toyota, T.: The effect of sea-ice growth on air–sea $CO_2$
flux in a tank experiment, Tellus, Ser. B, 58, 418–426, 2006.

Nomura, D., Inoue, H. Y., Toyota, T., and Shirasawa, K.: Effects of snow, snowmelting
and refreezing processes on air–sea-ice $CO_2$ flux, J. Glaciol., 56, 196, 262–270, 2010a.

Nomura, D., Eicken, H., Gradinger, R., and Shirasawa, K.: Rapid physically driven
inversion of the air-sea ice $CO_2$ flux in the seasonal landfast ice off Barrow, Alaska
after onset of surface melt, Cont. Shelf Res., 30, 1998–2004, 2010b.

Nomura, D., Granskog, M. A., Assmy, P., Simizu, D., and Hashida, G.: Arctic and
Antarctic sea ice acts as a sink for atmospheric $CO_2$ during periods of snow melt and
surface flooding, J. Geophys. Res. Oceans, 118, 6511–6524, 2013.



Merkouriadi, I., Gallet, J.-C., Graham, R. M., Liston, G. E., Polashenski, C., Rösel, A.,
and Gerland, S.: Winter snow conditions on Arctic sea ice north of Svalbard during the
Norwegian young sea ICE (N-ICE2015) expedition, J. Geophys. Res. Atmos., 122,
doi:10.1002/2017JD026753, 2017.
Papadimitriou, S., Kennedy, H., Norman, L., Kennedy, D. P., Dieckmann, G. S., and
co-authors: The effect of biological activity, $CaCO_3$ mineral dynamics, and $CO_2$
degassing in the inorganic carbon cycle in sea ice in late winter-early spring in the
Weddell Sea, Antarctica, J. Geophys. Res. 117, C08011, doi:10.1029/2012JC008058,
681 2012.
Papakyriakou, T., and Miller, L. A.: Springtime $CO_2$ exchange over seasonal sea ice in
the Canadian Arctic Archipelago, Ann. Glaciol., 52, 57, 215–224, 2011.
Parmentier, F. J. W., Christensen, T. R., Sørensen, L. L., Rysgaard, S., McGuire, A. D.,
and co-authors: The impact of lower sea-ice extent on Arctic greenhouse-gas exchange,
Nature Climate Change, 3, 195–202, doi:10.1038/nclimate1784, 2013.
Pierrot, D., Lewis, E. and Wallace, D. W. R.: MS Excel Program Developed for $CO_2$
System Calculations, ORNL/CDIAC-105a. Carbon Dioxide Information Analysis
Center, Oak Ridge National Laboratory, U.S. Department of Energy, Oak Ridge,
Tennessee, doi: 10.3334/CDIAC/otg.CO2SYS_XLS_CDIAC105a, 2006.
Powers, D., Oneill, K., and Colbeck, S. C.: Theory of natural convection in snow, J.
Geophys. Res.-Atmos., 90, 10641–10649, doi:10.1029/Jd090id06p10641, 1985.
Pringle, D. J., Miner, J. E., Eicken, H., and Golden, K. M.: Pore space percolation in sea
ice single crystals, J. Geophys. Res., 114, C12017, doi:10.1029/2008JC005145, 2009.
Rysgaard, S., Søgaard, D. H., Cooper, M., Pucko, M., Lennert, K., and co-authors:
Ikaite crystal distribution in winter sea ice and implications for $CO_2$ system dynamics,
The Cryosphere, 7, 707–718, doi:10.5194/tc-7-707-2013, 2013.




Rösel, A., Polashenski, C. M., Liston, G. E., King, J. A., Nicolaus, M. and co-authors:
N-ICE2015 snow depth data with Magna Probe (Data set), Norwegian Polar Institute,
doi:10.21334/npolar.2016.3d72756d, 2016a.

Rösel, A., Divine, D., King, J. A., Nicolaus, M., Spreen, G., and co-authors: N-ICE2015
total (snow and ice) thickness data from EM31 (Data set), Norwegian Polar Institute,
doi:10.21334/npolar.2016.70352512, 2016b.

Schindlbacher, A., Zechmeister-Boltenstern, S., Glatzel, G., and Jandl R.: Winter soil
respiration from an Austrian mountain forest, Agric, For. Meteorol., 146, 205–215,
doi:10.1016/j.agrformet.2007.06.001, 2007.

Severinghaus, J. P., Albert, M. R., Courville, Z. R., Fahnestock, M. A., Kawamura, K.,
and co-authors: Deep air convection in the firn at a zero-accumulation site, central
Antarctica, Earth Planet. Sci. Lett., 293, 359–367, doi:10.1016/J.Epsl.2010.03.003,

720    2010.


Stroeve, J. C., Serreze, M. C., Holland, M. M., Kay, J. E., Maslanik, J., and Barrett, A.
P.: The Arctic's rapidly shrinking sea ice cover: a research synthesis, Climatic Change,
110, 1005, doi:10.1007/s10584-011-0101-1, 2012.

Sørensen, L. L, Jensen, B., Glud, R. N., McGinnis, D. F., and Sejr, M. K.:
Parameterization of atmosphere-surface exchange of $CO_2$ over sea ice, The Cryosphere,
8: 853–866. doi:10.5194/tc-8-853-2014, 2014.

Takagi, K., Nomura, M., Ashiya, D., Takahashi, H., Sasa, K., and co-authors: Dynamic
carbon dioxide exchange through snowpack by wind-driven mass transfer in a conifer-
broadleaf mixed forest in northernmost Japan, Global Biogeochem. Cycles, 19, GB2012,
doi:10.1029/2004GB002272, 2005.



Toyota, T., Massom, R., Tateyama, K., Tamura, T., and Fraser, A.: Properties of snow
overlying the sea ice off East Antarctica in late winter 2007, Deep Sea Res. II, 58,
1137–1148, 2011.

Vancoppenolle, M., Meiners, K. M., Michel, C., Bopp, L., Brabant, F., and co-authors:
Role of sea ice in global biogeochemical cycles: emerging views and challenges, Quat.
Sci. Rev., 79, 207–230, 2013.

Weiss, R. F.: Carbon dioxide in water and seawater: the solubility of a non-ideal gas,
Mar. Chem., 2, 203–215, 1974.

Zemmelink, H. J., Delille, B., Tison, J.-L., Hintsa, E. J., Houghton, L., and co-authors:.
$CO_2$ deposition over the multi-year ice of the western Weddell Sea, Geophys. Res. Lett.,
33, L13606, doi:10.1029/2006GL026320, 2006.

Zemmelink, H. J., Dacey, J. W. H., Houghton, L., Hintsa. E. J., and Liss, P. S.:
Dimethylsulfide emissions over the multi-year ice of the western Weddell Sea, Geophys.
Res. Lett., 35, L06603, doi:10.1029/2007GL031847, 2008.

Zhou, J., Delille, B., Eicken, H., Vancoppenolle, M., Brabant, F., and co-authors:
Physical and biogeochemical properties in landfast sea ice (Barrow, Alaska): Insights
on brine and gas dynamics across seasons, J. Geophys. Res. 118, 6, 3172–3189, 2013.



**Figure captions**

Figure 1. Location map of the sampling area north of Svalbard during N-ICE2015.
Image of the sea ice concentrations (a) and station map (b) were derived from Special
Sensor Microwave Imager (SSM/I) satellite data for mean of February 2015 and from
Sentinel-1 (Synthetic Aperture Radar Sensor) satellite data, respectively.

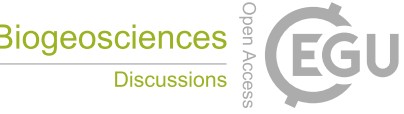

Figure 2. Photographs of the $CO_2$ flux chamber system at station YI1 north of Svalbard
on Friday 13 March 2015. $CO_2$ flux chamber was installed over the frost flower on the
new thin ice in the refreezing lead.

Figure 3. Time series of air temperature measured at the weather mast over the ice floe
(10 m height). Blank period indicates no data. Colored symbols indicate the date for the
chamber flux measurements. The horizontal dashed line indicates air temperature = 0°C.

Figure 4. Vertical profiles of temperature (a) and salinity (b) in snow and sea ice (top 20
cm). The horizontal line indicates snow–ice interface. Shaded area indicates sea ice. For
stations FI7 and YI2 and 3, we have no salinity data.

Figure 5. Relationships between mean air–sea ice $CO_2$ fluxes and temperature
difference between ice ($T_{ice}$) and atmosphere ($T_a$) (circle) and ice temperature (Tice)
(top 20 cm) (cross) for $F_{snow}$ (blue), $F_{ff}$ (green) and $F_{ice}$ (red) for young and first-year sea
ice.



**Table captions**

Table 1. Station, date for $CO_2$ flux measurement, position, floe number, surface
condition, ice type and thickness of snow, frost flowers, and sea ice.

a. Sea ice coring and snow sampling was conducted on 5 March 2015.

b. Sea ice coring and snow sampling was conducted on 10 March 2015.


Table 2. Station, snow density and water equivalent, brine volume fraction, and
temperature for sea ice (top 20 cm), brine temperature, salinity, DIC, TA, pCO$_2$ (pCO$_2$
$_b$), and atmospheric temperature, wind speed and pCO$_2$ (pCO$_2$ $_a$)[a].






a. $pCO_{2\,a}$ (µatm) was calculated from $CO_2$ concentration (ppmv) at Ny-Ålesund,
Svalbard (http://www.esrl.noaa.gov/gmd/dv/iadv/) taking into account saturated water
vapor and atmospheric pressure during sampling day.

b. Mean values for snow column.

c. "−" indicates no data.


Table 3. $CO_2$ flux measured over the snow ($F_{snow}$), frost flower ($F_{ff}$), and ice surface
($F_{ice}$). Values measured directly over undisturbed surfaces (either with frost flowers or
on snow surface) at a given station are indicated in bold.

a. Data of first $CO_2$ flux measurement after removal of snow or frost flower.

b. "−" means no data.

c. Data of OI1 was not included.





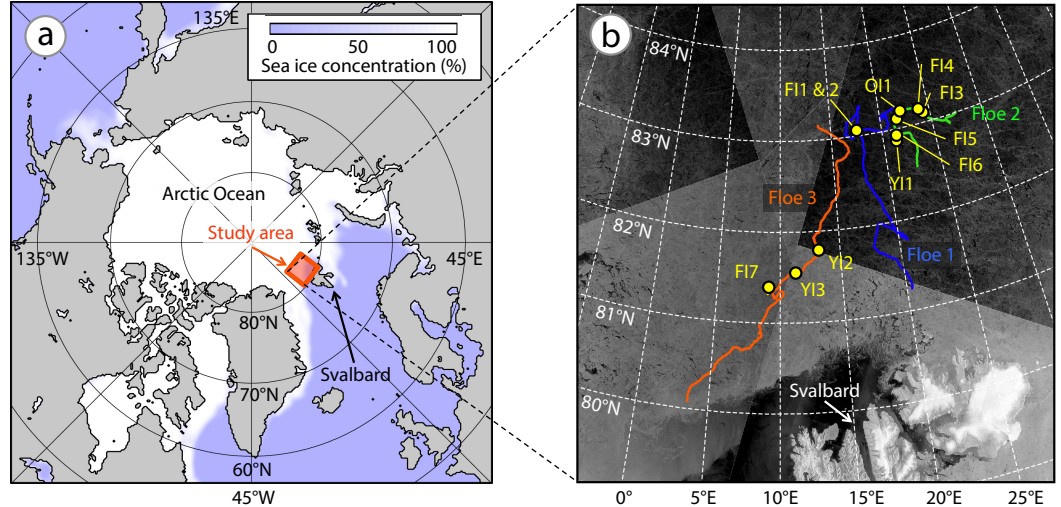



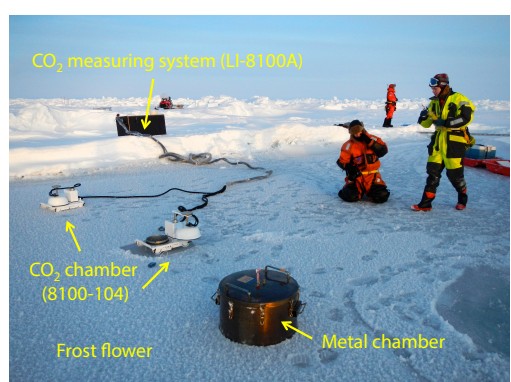

Nomura et al., Figuss. 2 (Single column size)





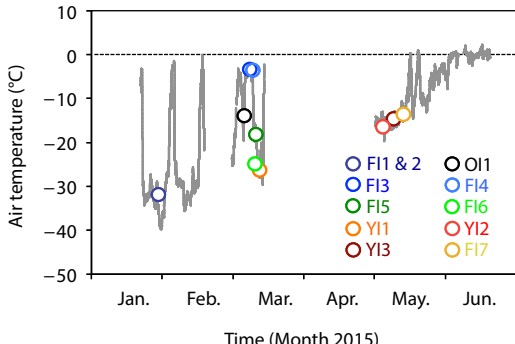





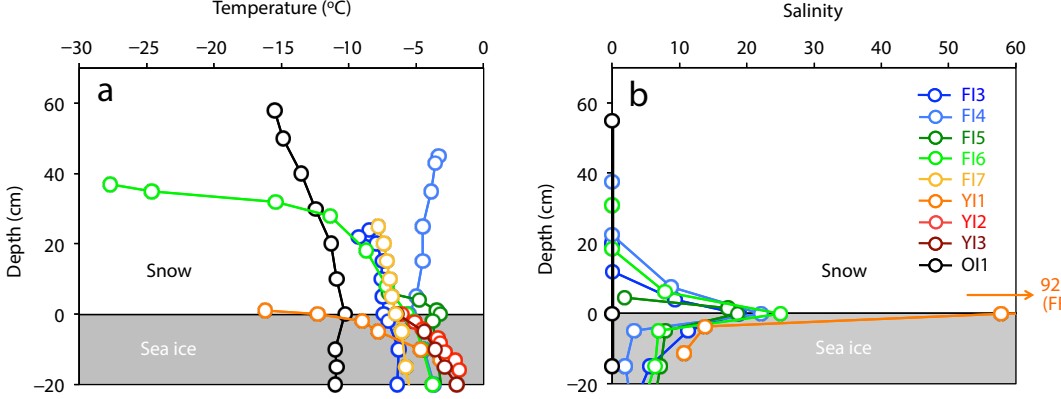

Nomura et al., Figure 4 (Double column size)





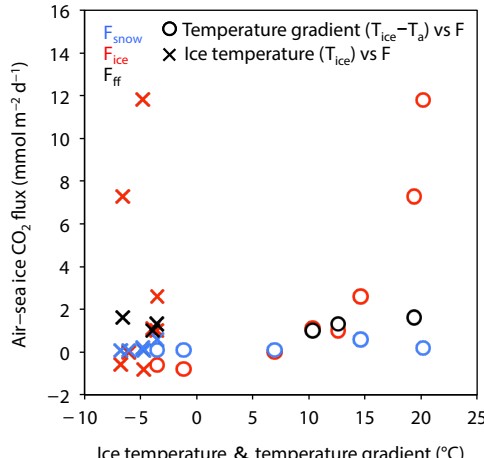

Nomura et al., Figure 5 (Single column size)



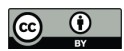

Table 1. Station, position, date for $CO_2$ flux measurement, floe number, surface condition, ice type and thickness of snow, frost flower, and sea ice.

| Station | Position | Date of 2015 | Floe number | Surface condition | Ice type[c] | Thickness (cm) | | |
|---|---|---|---|---|---|---|---|---|
| | | | | | | Snow | Frost flower | Sea ice |
| FI1 | 83°03.77N, 17°34.94E | 28 January | 1 | Frost flower | First-year ice | 0.0 | 1.0 | 37.0 |
| FI2 | 83°03.77N, 17°34.94E | 28 January | 1 | Snow | First-year ice | 8.0 | No | 35.0 |
| FI3 | 83°08.00N, 24°09.02E | 5 and 8 March[a] | 2 | Snow | First-year ice | 29.0 | No | 98.0 |
| FI4 | 83°10.56N, 22°09.42E | 9 March | 2 | Snow | First-year ice | 36.0 | No | 92.0 |
| FI5 | 83°06.02N, 21°38.29E | 10 and 11 March[b] | 2 | Snow | First-year ice | 3.0 | No | 48.0 |
| FI6 | 82°55.36N, 21°25.92E | 12 March | 2 | Snow | First-year ice | 37.0 | No | 69.0 |
| FI7 | 81°22.18N, 08°59.93E | 13 May | 3 | Snow | First-year ice | 26.5 | No | 127.0 |
| YI1 | 82°52.52N, 21°16.54E | 13 March | 2 | Frost flower | Young ice | 0.0 | 1.0 | 15.0 |
| YI2 | 81°46.53N, 13°16.00E | 5 May | 3 | Snow and frost flower mixed | Young ice | 2.5 | 2.5 | 17.5 |
| YI3 | 81°32.45N, 11°17.20E | 9 May | 3 | Snow and frost flower mixed | Young ice | 2.0 | 2.0 | 22.0 |
| OI1 | 83°07.18N, 24°25.59E | 6 March | 2 | Snow | Old ice (multi-year ice) | 60.0 | No | >200 |

a. Sea ice coring, brine and snow sampling was conducted on 5 March 2015.
b. Sea ice coring, brine and snow sampling was conducted on 10 March 2015.
c. Ice type was categorized based on WMO (1970).





Table 2. Station, snow density and water equivalent, brine volume fraction and temperature for sea ice (top 20 cm), brine temperature, salinity, DIC, TA, pCO$_2$ (pCO$_{2b}$) and atmospheric temperature, wind speed, and pCO$_2$ (pCO$_{2a}$)[a].

| Station | Snow | | Sea ice (top 20 cm) | | Brine | | | | | Atmosphere | | |
| | Density[b] (kg m$^{-3}$) | Water equivalent (kg m$^{-2}$) | Brine volume fraction (%) | Temperature (°C) | Temperature (°C) | Salinity | DIC (µmol kg$^{-1}$) | TA (µmol kg$^{-1}$) | pCO$_{2b}$ (µatm) | Temperature (°C) | Wind speed (m second$^{-1}$) | pCO$_{2a}$ (µatm) |
|---|---|---|---|---|---|---|---|---|---|---|---|---|
| FI1 | –[c] | –[c] | –[c] | –[c] | –[c] | –[c] | –[c] | –[c] | –[c] | –31.6 | 4.0 | 405 |
| FI2 | –[c] | –[c] | –[c] | –[c] | –[c] | –[c] | –[c] | –[c] | –[c] | –31.6 | 4.0 | 405 |
| FI3 | 399 | 104 | 9 | –6.8 | –5.2 | 84.8 | 4628 | 5539 | 427 | –3.3 | 9.0 | 400 |
| FI4 | 400 | 180 | 9 | –4.7 | –5.3 | 86.6 | 4433 | 5490 | 334 | –3.5 | 6.2 | 386 |
| FI5 | 268 | 11 | 17 | –3.5 | –3.3 | 51.8 | 3261 | 3518 | 472 | –18.1 | 6.8 | 389 |
| FI6 | 343 | 127 | 13 | –4.8 | –4.8 | 84.0 | 4841 | 5493 | 693 | –25.0 | 3.6 | 400 |
| FI7 | –[c] | –[c] | –[c] | –6.1 | –[c] | –[c] | –[c] | –[c] | –[c] | –13.0 | 5.8 | 405 |
| YI1 | –[c] | –[c] | 17 | –6.6 | –[c] | –[c] | –[c] | –[c] | –[c] | –26.0 | 2.6 | 402 |
| YI2 | –[c] | –[c] | –[c] | –3.6 | –[c] | –[c] | –[c] | –[c] | –[c] | –16.2 | 4.5 | 407 |
| YI3 | –[c] | –[c] | –[c] | –3.9 | –[c] | –[c] | –[c] | –[c] | –[c] | –14.2 | 6.7 | 410 |
| OI1 | –[c] | –[c] | 0 | –10.8 | –[c] | –[c] | –[c] | –[c] | –[c] | –13.5 | 4.7 | 397 |

a. pCO$_{2a}$ (µatm) was calculated from CO$_2$ concentraion (ppmv) at Ny-Ålesund, Svalbard (http://www.esrl.noaa.gov/gmd/dv/iadv/) taking into account the saturated water vapor and atmospheric pressures at sampling day.

b. Mean values for column.

c. "–" indicates no data.





Table 3. CO$_2$ flux measured over the snow (F$_{snow}$), frost flower (F$_{ff}$) and ice surface (F$_{ice}$).

| Station | CO$_2$ flux (mmol C m$^{-2}$ day$^{-1}$) (mean ± 1SD) (number of measurement) | | |
| | Natural flux | | Potential flux |
| | F$_{snow}$ | F$_{ff}$ | F$_{ice}$ [a] |
| --- | --- | --- | --- |
| FI1 | –[a] | +0.1 ± 0.1 (n=7) | –[b] |
| FI2 | +0.4 ± 0.3 (n=13) | –[b] | –[b] |
| FI3 | +0.1 ± 0.1 (n=7) | –[b] | -0.6 |
| FI4 | +0.1 ± 0.1 (n=6) | –[b] | -0.8 |
| FI5 | +0.6 ± 0.3 (n=5) | –[b] | +2.6 |
| FI6 | +0.2 ± 0.1 (n=5) | –[b] | +11.8 |
| FI7 | +0.1 ± 0.1 (n=10) | –[b] | ±0.0 |
| YI1 | –[b] | +1.6 ± 0.2 (n=6) | +7.3 |
| YI2 | –[b] | +1.3 ± 0.2 (n=9) | +1.0 |
| YI3 | –[b] | +1.0 ± 0.4 (n=8) | +1.1 |
| OI1 | +0.1 ± 0.0 (n=6) | –[b] | +0.2 |
| Mean[c] | +0.2 ± 0.2 (n=46) | +1.0 ± 0.6 (n=30) | +2.5 ± 4.3 (n=9) |

a. Data of first measurement after removal of snow or frost flower.

b. "–" means no data.

c. Data of OI1 was not included.