# Peer review of "CO2 flux over young and snow-covered Arctic sea ice in"

_Biogeosciences, 2017_

## Referee Comment (RC1) · Anonymous Referee #1 · 11 Feb 2018

General comments

Nomura et al present an interesting analysis of rare data capturing CO2 fluxes between sea ice and the atmosphere in Arctic winter, spring and summer as part of the N-ICE project. The methods are robust, the data are of high quality and significant value, and the arguments laid out in the paper will be of wide interest amongst the sea ice and CO2 communities.

However, the manuscript comes across as a little rushed in its current form, and I believe it would be improved significantly by adding more detail and explaining more clearly the key points. I recommend acceptance for publication after moderate revisions.

[Figure]

Results are presented in summary tables. In general, I find that not enough information is presented for the reader to easily follow the arguments made in the paper, and I think some may even be misleading. For instance, based on Table 3, you argue that Fice is greater than Fsnow and thus make the argument that snow cover reduces flux magnitude. From the table, it appears that this is only demonstrably true for two out of the seven first-year ice stations. Two of the stations appear to have negative fluxes, but this is not addressed in the text at all, but seems to me to be quite important. These factors should be discussed in much greater detail in the text. Given the variability in your results, I think it is necessary to present the actual data, rather than just summary data. This would probably be best as figures, to accompany the summary tables. On a similar note, you have the number of measurements listed for F-snow and F-ff in table 3, but why not F-ice. Please include this information and error estimates.

It is also quite difficult in general to follow the flow through and between the different tables, for example discussion of the relationship between flux magnitude and snow thickness or water equivalent. The text needs more detail to guide the reader's understanding and some more figures would certainly help.

Specific comments

Introduction: it would be useful to include a little more information about what we know about ice-atmosphere $CO_2$ fluxes in the context of ocean-atmosphere fluxes overall in the Arctic, and how they may change in the future. That would set the scene nicely for your statements at the end about ice-atmosphere fluxes being important in the context of a changing Arctic and the broader implications of your work. The final paragraph (line 107) could also be much stronger and punchier.

Line 125-127: state specifically which stations you are referring to. I assume "young ice", but this should be explicit. That might also help the descriptions of relationships between variables in the discussion, as mentioned in "general comments".

Line 155-157: does this not contradict your argument that snow provides insulation?
Perhaps it would help to mention timescales of T change/stability.

Line 162: I think you have air and ice surface the wrong way round.

Line 172: I think you should distinguish between stations where snow was cleared and where the sea ice surface was naturally snow-free. Given your arguments about the effects of snow cover, I assume this is significant.

Line 185-187: clarify when temperature was measured.

Line 192-193: why was carbonate chemistry only measured at these four stations? This should be explained. It also means that table 2 looks like there is a lot of data missing; perhaps there is a better way to present these data?

Line 220: I think this should be Guildline PORTASAL salinometer Model8410A

Line 239-240 and 239-250: this strongly suggests that the constants are not valid for your conditions. The following clearly attempts to justify its use, but it is not clear why the 40 % uncertainty does not apply to your data, which would mean that none of your calculated values would have statistically significant differences. Please clarify.

Line 253-254: please give enough information for the reader to understand this calculation, without having to dig out an old reference.

Methods: please include information about how atmospheric pCO2 was measured. It comes later as a footnote to a table, but should be included here.

Line 275-276: state which stations you are referring to. This would help in general in various places in the text.

Line 279-280: I think it would help to demonstrate this point if you plotted air temperature on figure 4, so that the relation is clear.

Line 285-286: can you highlight on figure 4b which measurements are from frost flowers?

Line 292 and table 2: you present data from the top 20 cm, which presumably means your top two 10cm slices. Why do you only present the top 20cm when most cores are longer? Would it be better to present profiles to show downcore variability? If not, please justify presenting only the top 20 cm and provide error/uncertainty estimates from averaging of values from two core slices.

Line 322: "except for station OI1". Should this also say YI1 as it does in section 3.2?

Line 324: "..and in cases the thick insulating snow cover". Does not make sense. In certain cases? In cases where...?

Line 355-358: this statement is only true for FI5, FI6 and YI1. Same comment for line 372-373.

Line 357: Where you state that one value or group of values is lower than another, please provide relevant statistical details (e.g. t-test, z-test etc.)

Line 372-382: This paragraph is an example of where a lot more detail is required to demonstrate your points. Flux direction, magnitude and relationships between variables all need to be discussed for the different stations.

Line 380: reference to table 3. You need to be specific about what you are referring to that shows that flux is reduced by the presence of snow. If you compare FI5 and FI6, FI6 shows a much greater potential flux but actually has a greater snow thickness and water equivalent than FI5. This should be incorporated into your comparisons.

Line 396-399: How will footprint size make such a big difference? If it arises from small-scale heterogeneity in time and/or space, this should be stated. Are there any other reasons worthy of mention?

Line 401-406: your fluxes are at the lower end of positive values – this should be stated, and elaborated on to discuss negative fluxes as well as positive ones (as per my earlier comment).

Line 406: should be "up to +11.8" or somehow make it clear that this is the maximum value.

Line 432-461: this section emphasises the importance of the temperature gradient in modifying fluxes and gives the impression that this is the most important variable. In fact, the correlation between temperature difference and flux is less strong than the correlation with pCO2 difference between the ice and atmosphere (given in line 310). This would be much clearer and more reflective of what the data show, if both variables were discussed here in terms of their relative importance overall and such a strong emphasis on temperature dampened. I also think it would help to add to figure 5 a panel which plots pCO2 difference vs. flux, to show the two relationships directly.

Line 458-459: "for young sea ice likely the frost flower conditions". Does not make sense.

Line 468-473: from the data presented in table 3, not all stations can be described as showing CO2 sources. Some clearly show sink behaviour (negative fluxes), and for a number of others, the uncertainty on flux estimates cannot confidently be described as a source, e.g. when flux = $0.1\pm0.1$. This is particularly the case given that you state the detection limit as 0.1. This also needs to be considered in your discussion.

Line 476-477: This should not be presented as a conclusion.

Line 485-488: I think you undersell the importance of your work here, and you could make more compelling statements about the role of sea ice in CO2 fluxes in a changing Arctic.

Figure 3. should this cite Hudson et al., 2015?

Table 2. Consider adding an extra column for $\Delta$pCO2 (air-sea difference) to aid understanding.

Table 3. The key thing that jumps out for me is that natural flux is much higher for frost flowers than snow. I would have thought that's worth highlighting in your discussion.

Technical corrections

In general, the manuscript is well-written, the technical language is appropriate, and the standard of English is good. However, there are a couple of points to check throughout the text: use of the definite/indefinite article; singular/plural nouns and their following verbs e.g. frost flowers, was/were.

Line 84: should be transport by molecular diffusion

Line 218: remove hyphens

Line 377: I think 0.0 is a mistake.

Table 3: brackets in the top line are confusing.

[Figure]

---

## Referee Comment (RC2) · Anonymous Referee #2 · 16 Feb 2018

The manuscript makes interesting observations of CO2 flux through sea ice, but requires extensive improvement. It was never articulated how this study is novel. I feel that it perhaps may be novel, but it is unclear how in its current form. Major revisions are needed before this manuscript can be considered publishable.

The abstract is borderline uninformative. What are characteristic fluxes? Are these important? Of course CO2 can flux through sea ice, but it's hard for the reader to gage exactly how trivial this is without values in the abstract to justify reading the rest of the paper.

The sentence beginning line 61 is a reference dump. What did these studies find and how does it build to the importance (or lack thereof) of the present manuscript?

[Figure]

68: Sea-ice CO_2 fluxes

On line 81, please see Massman et al. (1995) as the fundamental reference on this topic (https://www.fs.fed.us/rm/pubs_exp_for/glees/exp_for_glees_1995_massman.pdf).

Somewhat harsh transition before the last paragraph of the introduction. Please state more clearly how the background materials presented tie directly to the proposed study and therefore what makes the present study novel. Material in section 4.3 could help. (note that there are also many reference dumps here. Please explain what the studies found; it is your job to make the reader's job easy (https://www.sesync.org/blog/the-writers-job).

on line 132, how was it ensured that placement of chambers did not perturb the pressure gradients in the snow? Creating pressure gradients can push CO2 out (or pull it in).

on 144, please see Bain et al. (2005) as a relevant reference for wind-induced effects (https://www.sciencedirect.com/science/article/pii/S0168192305001164)

Frost flowers are first introduced in the paragraph beginning on line 146. One assumes that these are somehow important for CO2 flux? The notion was not previously introduced. (see line 360. This belongs in the intro). I agree with Reviewer 1 that the manuscript was prepared somewhat hastily.

153: what is station FI6? Abbreviations are introduced before they are explained. It would help to explain the geography of the site before the measurements, also to ensure that measurements were made with a random design in mind.

Extensive English improvement is needed in section 2.3

On line 266, what does 'near-constant 0 C' mean?

60.0 cm sounds rather specific for a measurement of snow which I assume has frequent small undulations, either at the snow surface or snow-ice interface

in section 3.4, per day is not a SI unit, and diurnal patterns in the flux may make it difficult to scale from the native measurements (in the SI units of seconds) to the full day.

416: the abbreviation F was introduced far earlier.

432: this is actually interesting. By focusing on the challenge of estimating gas transfer velocity, the manuscript has some novel features. These might be initial hypotheses for future work if causality can't be determined, but the mechanisms of sea ice/atmosphere gas exchange make for a more interesting analysis even if remaining questions are left.

Figure 4: avoid simultaneous use of red and green in a figure.

---

## Author Comment (AC1) · 27 Mar 2018

**Point-by-point responses to Review #1 and 2.**

Journal: BG
Title: $CO_2$ flux over young and snow-covered Arctic sea ice in winter and spring
Author (s): Daiki Nomura et al.
MS No.: bg-2017-521
MS Type: Research article

We thank the reviewers for their valuable comments, which have helped us to improve the manuscript.

For clarity, the authors' responses are inserted as green text.

**Anonymous Referee #1**

General comments

Nomura et al present an interesting analysis of rare data capturing CO2 fluxes between sea ice and the atmosphere in Arctic winter, spring and summer as part of the N-ICE project. The methods are robust, the data are of high quality and significant value, and the arguments laid out in the paper will be of wide interest amongst the sea ice and CO2 communities. However, the manuscript comes across as a little rushed in its current form, and I believe it would be improved significantly by adding more detail and explaining more clearly the key points. I recommend acceptance for publication after moderate revisions. Results are presented in summary tables. In general, I find that not enough information is presented for the reader to easily follow the arguments made in the paper, and I think some may even be misleading. For instance, based on Table 3, you argue that Fice is greater than Fsnow and thus make the argument that snow cover reduces flux magnitude. From the table, it appears that this is only demonstrably true for two out of the seven first-year ice stations. Two of the stations appear to have negative fluxes, but this is not addressed in the text at all, but seems to me to be quite important. These factors should be discussed in much greater detail in the text. Given the variability in your results, I think it is necessary to present the actual data, rather than just summary data. This would probably be best as figures, to accompany the summary tables. On a similar note, you have the number of measurements listed for F-snow and F-ff in table 3, but why not F-ice. Please include this information and error estimates. It is also quite difficult in general to follow the flow through and between the different tables, for example discussion of the relationship between flux magnitude and snow thickness or water equivalent. The text needs more detail to guide the reader's under-standing and some more figures would certainly help.

We are grateful for your favorable assessment. We have made changes in response to all of your recommendations and edited the text improve the readability of the text.

Now we have indicated the stations for each result (e.g., for stations FI5 and FI6) in the text. In addition, we have added following information about the negative fluxes and reason for single $F_{ice}$ measurement in the text:

"For $F_{ice}$, there were negative $CO_2$ fluxes at stations FI3 and FI4 (–0.6 mmol C m$^{-2}$ day$^{-1}$ for FI3 and –0.8 mmol C m$^{-2}$ day$^{-1}$ for FI4) (Table 3). These fluxes corresponded to low or negative $\Delta pCO_{2\ b-a}$ as compared to that in atmosphere (Table 2 and Figure 6). Negative $CO_2$ fluxes should correspond to negative $\Delta pCO_{2\ b-a}$. Therefore, the uncertainty for the calculation of carbonate chemistry may be one reason for the discrepancy in $pCO_2$ calculation in these conditions (Brown et al., 2014)."

"During first $CO_2$ flux measurements (about 30 minutes), ice surface temperature was stable at –5.8°C, suggesting that the effect of removing snow on the variation of sea ice surface temperature was negligible within 30 minutes. The ice surface temperature decreased from –5.8°C to –8.0°C at 200 minutes after removal of snow. Therefore, in this paper, the data of the initial 30 minutes of $CO_2$ flux measurement after removal of snow or frost flowers was used."

In order to present actual data, we have added relationships between $pCO_2$ and $CO_2$ flux in figure showing the relationships between temperature and $CO_2$ flux (Figure 6). In addition, we have made new figure showing the temporal variation of $CO_2$ concentration within chamber (Figure 3).

Specific comments

Introduction: it would be useful to include a little more information about what we know about ice-atmosphere CO2 fluxes in the context of ocean-atmosphere fluxes overall in the Arctic, and how they may change in the future. That would set the scene nicely for your statements at the end about ice-atmosphere fluxes being important in the context of a changing Arctic and the broader implications of your work. The final paragraph (line 107) could also be much stronger and punchier.

Thank you for your suggestions.

We have now added some more discussion on the results from other work in the Arctic, and to emphasize the lack of observations in the pack ice:

"In the ice covered Arctic Ocean, storm periods, with high wind speeds and open leads are important for air-to-sea $CO_2$ fluxes (Fransson et al., 2017), due to the under-saturation of the surface waters in $CO_2$ with respect to the atmosphere. On the other hand, the subsequent ice growth and frost flowers formation in these leads promote ice-to-air $CO_2$ fluxes in winter (e.g. Barber et al., 2014). Given the fact that Arctic sea ice is shrinking and shifting from multi-year ice to first-year ice, the area of open ocean and thinner seasonal ice is increasing. Therefore, the contribution of open ocean/thinner sea ice surface to the overall $CO_2$ fluxes of the Arctic Ocean is potentially increasing. However, due to the difficulty in acquiring observations over the winter pack ice, most of the winter $CO_2$ flux measurements were examined over the Arctic landfast ice. Therefore, there is a definite lack of information on conditions during wintertime, especially from Arctic pack ice." in introduction.

"Rare $CO_2$ flux measurements from Arctic pack ice show that two types of ice are significant contributors to the release of $CO_2$ from ice to the atmosphere during winter and spring: young thin ice with thin layer of snow, and old (several weeks) snow covered thick ice." in abstract.

We have changed from "Arctic sea ice" to "Arctic pack ice" in title.

To emphasize the novelty of our work, we have rewritten the final paragraph;

"The Norwegian young sea ICE (N-ICE2015) campaign in winter and spring 2015 provided opportunities to examine $CO_2$ fluxes between sea ice and atmosphere in a variety of snow and ice conditions in pack ice north of Svalbard. Formation of leads and their rapid refreezing allowed us to examine air–sea ice $CO_2$ fluxes over thin young sea ice, occasionally covered with frost flowers in addition to the snow-covered older ice that covers most of the pack ice area. The objectives of this study were to understand the effects of i) thin sea ice and frost flowers formations on the air–sea ice $CO_2$ flux in leads, ii) effect of snow-cover on the air–sea ice $CO_2$ flux over thin, young ice in the Arctic Ocean during winter and spring seasons, and iii) of the effect of the temperature difference between sea ice and atmosphere (including snow cover) on the air–sea ice $CO_2$ flux. ".

Line 125-127: state specifically which stations you are referring to. I assume "young ice", but this should be explicit. That might also help the descriptions of relationships between variables in the discussion, as mentioned in "general comments".

Thank you for this suggestion. We have added the specific information for station "station YI1". For the descriptions of relationships between variables in the discussion, please see your general comments.

Line 155-157: does this not contradict your argument that snow provides insulation? Perhaps it would help to mention timescales of T change/stability.

We agree with your comments. We have added:

"The ice surface temperature decreased from $-5.8°C$ to $-8.0°C$ at 200 minutes after removal of snow. Therefore, in this paper, the data of the initial 30 minutes of $CO_2$ flux measurement after removal of snow or frost flowers was used." in the text.

Line 162: I think you have air and ice surface the wrong way round.

Correct, well spotted. We have corrected.

Line 172: I think you should distinguish between stations where snow was cleared and where the sea ice surface was naturally snow-free. Given your arguments about the effects of snow cover, I assume this is significant.

We have no station where the sea ice surface was naturally snow-free (unless frost flowers are not considered as snow) (Table 1).

Line 185-187: clarify when temperature was measured.

We have added "during $CO_2$ flux measurements (approximately 60 minutes after the onset of the $CO_2$ flux measurement)" in the text.

Line 192-193: why was carbonate chemistry only measured at these four stations? This should be explained. It also means that table 2 looks like there is a lot of data missing; perhaps there is a better way to present these data?

At some occasions there was simply no time to collect the samples right after the flux measurements were taken, due to diverse and challenging conditions in the field. Due to the technical reason, we could not obtain the brine, except for four stations. Therefore, we have no samples for brine carbonate chemistry, except for four stations. We have added "Due to technical reason, data of snow, sea ice, and brine data were not obtained" in Table 2 caption.

Line 220: I think this should be Guildline PORTASAL salinometer Model8410A

Correct. Changed accordingly.

Line 239-240 and 239-250: this strongly suggests that the constants are not valid for your conditions. The following clearly attempts to justify its use, but it is not clear why the 40% uncertainty does not apply to your data, which would mean that none of your calculated values would have statistically significant differences. Please clarify.

For $F_{ice}$, there was negative $CO_2$ flux for stations FI3 although $\Delta pCO_{2\,b-a}$ was positive. Negative $CO_2$ fluxes should correspond to negative $\Delta pCO_{2\,b-a}$. Therefore, the uncertainty for the calculation of carbonate chemistry may be one reason for the discrepancy in $pCO_2$ calculation in these conditions (Brown et al., 2014).".

We have added "For $F_{ice}$, there were negative $CO_2$ fluxes at stations FI3 and FI4 (–0.6 mmol C m$^{-2}$ day$^{-1}$ for FI3 and –0.8 mmol C m$^{-2}$ day$^{-1}$ for FI4) (Table 3). These fluxes corresponded to low or negative $\Delta pCO_{2\,b-a}$ as compared to that in atmosphere (Table 2 and Figure 6). Negative $CO_2$ fluxes should correspond to negative $\Delta pCO_{2\,b-a}$. Therefore, the uncertainty for the calculation of carbonate chemistry may be one reason for the discrepancy in $pCO_2$ calculation in these conditions (Brown et al., 2014).".

Line 253-254: please give enough information for the reader to understand this calculation, without having to dig out an old reference.

We have added newer reference "Petrich and Eicken, 2010". This is a rather standard method for sea-ice, thus we would not like to use space to explain the derivation of porosity in more detail than referring to the source.

Petrich, C. and Eicken, H.: Growth, structure and properties of sea ice, in Thomas, D. N. and Dieckmann, G. S. eds., Sea Ice, 2nd ed., Oxford, Wiley-Blackwell, 23–77, 2010.

Methods: please include information about how atmospheric pCO2 was measured. It comes later as a footnote to a table, but should be included here.

We agree with your comment. We have added "The $pCO_2$ of atmosphere was calculated from $CO_2$ concentration (ppmv) at Ny-Ålesund, Svalbard (http://www.esrl.noaa.gov/gmd/dv/iadv/) taking into account saturated water vapor and atmospheric pressure during sampling day." in the text.

Line 275-276: state which stations you are referring to. This would help in general in various places in the text.

Agree. We have added "at station YI1" in the text, and also in other locations in the text to make the reasoning easier to follow.

Line 279-280: I think it would help to demonstrate this point if you plotted air temperature on figure 4, so that the relation is clear.

We have added air temperature on Figure 5a.

Line 285-286: can you highlight on figure 4b which measurements are from frost flowers?

We have changed the range of salinity in Figure 5b and added arrow to indicate frost flower data.

Line 292 and table 2: you present data from the top 20 cm, which presumably means your top two 10cm slices. Why do you only present the top 20cm when most cores are longer? Would it be better to present profiles to show downcore variability? If not, please justify presenting only the top 20 cm and provide error/uncertainty estimates from averaging of values from two core slices.

We have used average temperature for top 20 cm sea ice because the environmental information at the top of sea ice were important parameters regulating the $CO_2$ flux at sea ice surface. Unlikely the conditions deeper down in the ice will be important for such a short period of measurement given fluxes in the ice would be diffusion driven. We have added the range of temperature at top 20 cm sea ice in Table 2.

Line 322: "except for station OI1". Should this also say YI1 as it does in section 3.2?

Correct. We have added YI1 in the text.

Line 324: "..and in cases the thick insulating snow cover". Does not make sense. In certain cases? In cases where. . .?

We agree with your comments. We have changed to ", except for station OI1 (Tables 1 and 2)".

Line 355-358: this statement is only true for FI5, FI6 and YI1. Same comment for line 372-373.

Correct. We have added ", especially for stations FI5 and FI6".

Line 357: Where you state that one value or group of values is lower than another, please provide relevant statistical details (e.g. t-test, z-test etc.)

We agree with your comments. We have deleted "mean" and added ", especially for stations FI5, FI6, and YI1." in the text.

Line 372-382: This paragraph is an example of where a lot more detail is required to demonstrate your points. Flux direction, magnitude and relationships between variables all need to be discussed for the different stations.

We have added information of flux direction, magnitude and relationships between variables ($F_{snow}$/$F_{ice}$ ratio and water equivalent) all need to be discussed for the different stations. New paragraph is:

"The magnitude of positive $F_{snow}$ is less than $F_{ice}$ for stations FI5 and FI6 (Table 3) indicating that the potential $CO_2$ flux from sea ice decreased due to the presence of snow. Previous studies have shown that snow accumulation over sea ice effectively impede $CO_2$ exchange (Nomura et al., 2013; Brown et al., 2015). Nomura et al. (2013) reported that 50–90% of the potential $CO_2$ flux was reduced due to the presence of snow/superimposed-ice at the water equivalent of 57–400 kg m$^{-2}$, indicating that the snow properties are an important factor that controls the $CO_2$ exchange through a snowpack. Comparisons between stations FI5 and FI6 for $F_{snow}$/$F_{ice}$ ratio (0.2 for FI5 and 0.0 for FI6) and water equivalent (11 kg m$^{-2}$ for FI5 and 127 kg m$^{-2}$ for FI6) indicate that the potential $CO_2$ flux is reduced (80% for FI5 and 98% for FI6 of the potential $CO_2$ flux) with increasing water equivalent. Although the magnitude of the potential $CO_2$ flux through the sea ice surface decreased by the presence of snow for stations FI5 and FI6 (Table 3), the snow surface still presents a $CO_2$ source to the atmosphere for low snow density and shallow depth conditions (e.g., +0.6 mmol C m$^{-2}$ day$^{-1}$ for FI5)."

Line 380: reference to table 3. You need to be specific about what you are referring to that shows that flux is reduced by the presence of snow. If you compare FI5 and FI6, FI6 shows a much greater potential flux but actually has a greater snow thickness and water equivalent than FI5. This should be incorporated into your comparisons.

We agree with your comments. We have added "for stations FI5 and FI6".

Line 396-399: How will footprint size make such a big difference? If it arises from small-scale heterogeneity in time and/or space, this should be stated. Are there any other reasons worthy of mention?

To clarify we have added the following "The eddy covariance method reflects a flux integrated over a large area, that can contain several different surface types. Therefore, eddy-covariance appears to be more useful for understanding fluxes at large special and temporal scales. On the other hand, the chamber method reflects the area where chamber was covered, and it is useful for understanding the relationship between fluxes and ice surface conditions on smaller scales. The different spatial scales of the two methods may be therefore one reason for the discrepancy in $CO_2$ flux measurements."

Line 401-406: your fluxes are at the lower end of positive values – this should be stated, and elaborated on to discuss negative fluxes as well as positive ones (as per my earlier comment).

We have added "of positive values".

We have added "For $F_{ice}$, there were negative $CO_2$ fluxes at stations FI3 and FI4 (–0.6 mmol C m$^{-2}$ day$^{-1}$ for FI3 and –0.8 mmol C m$^{-2}$ day$^{-1}$ for FI4) (Table 3). These fluxes corresponded to low or negative$\Delta pCO_2$ $_{b-a}$ as compared to that in atmosphere (Table 2 and Figure 6). Negative $CO_2$ fluxes should correspond to negative $\Delta pCO_2$ $_{b-a}$. Therefore, the uncertainty for the calculation of carbonate chemistry may be one reason for the discrepancy in $pCO_2$ calculation in these conditions (Brown et al., 2014)." in the text.

Line 406: should be "up to +11.8" or somehow make it clear that this is the maximum value.

We agree with your comments. We have added "up to".

Line 432-461: this section emphasises the importance of the temperature gradient in modifying fluxes and gives the impression that this is the most important variable. In fact, the correlation between temperature difference and flux is less strong than the correlation with pCO2 difference between the ice and atmosphere (given in line 310). This would be much clearer and more reflective of what the data show, if both variables were discussed here in terms of their relative importance overall and such a strong emphasis on temperature dampened. I also think it would help to add to figure 5 a panel which plots pCO2 difference vs. flux, to show the two relationships directly.

We agree with your comments. We indicated that both variables ($\Delta pCO_2$ $_{b-a}$ and temperature difference) affect $CO_2$ flux. For example, we compared our data (e.g. for station FI6) with a previous study (Nomura et al., 2006) for each variable. The $\Delta pCO_{2\,b-a}$ was similar (297 µatm for Nomura et al., 2006 and 293 µatm for FI6) while temperature difference was not same (4.5°C for Nomura et al., 2006 and 20.2°C for FI6). In addition, the $CO_2$ flux was +0.7 mmol C m$^{-2}$ day$^{-1}$ for Nomura et al., 2006 and +11.8 mmol C m$^{-2}$ day$^{-1}$ for FI6. These results suggested that temperature difference enhanced the $CO_2$ flux between sea ice and atmosphere at the same $\Delta pCO_{2\,b-a}$. On the other hand, the variation of $\Delta pCO_{2\,b-a}$ would be modified $CO_2$ flux as shown in equation ($F_{CO2} = r_b\,k\,\alpha\,\Delta pCO_{2\,b-a}$). For the relationships between $CO_2$ flux and $\Delta pCO_{2\,b-a}$ as indicated in section 3.4, $CO_2$ flux values included the effect of the temperature difference. Therefore, it is difficult to divide the relative importance for $\Delta pCO_{2\,b-a}$ and temperature difference.

We have added relationships between $pCO_2$ and $CO_2$ flux in figure showing the relationships between temperature and $CO_2$ flux (Figure 6).

Line 458-459: "for young sea ice likely the frost flower conditions". Does not make sense.

We agree with your comments. We have changed to "for young sea ice with frost flowers (e.g. station YI1)".

Line 468-473: from the data presented in table 3, not all stations can be described as showing CO2 sources. Some clearly show sink behaviour (negative fluxes), and for a number of others, the uncertainty on flux estimates cannot confidently be described as a source, e.g. when flux = 0.1±0.1. This is particularly the case given that you state the detection limit as 0.1. This also needs to be considered in your discussion.

We agree with your comments. We have added:

"For $F_{ice}$, there were negative $CO_2$ fluxes at stations FI3 and FI4 (–0.6 mmol C m$^{-2}$ day$^{-1}$ for FI3 and –0.8 mmol C m$^{-2}$ day$^{-1}$ for FI4) (Table 3). These fluxes corresponded to low or negative $\Delta pCO_{2\,b-a}$ as compared to that in atmosphere (Table 2 and Figure 6). Negative $CO_2$ fluxes should correspond to negative $\Delta pCO_{2\,b-a}$. Therefore, the uncertainty for the calculation of carbonate chemistry may be one reason for the discrepancy in $pCO_2$ calculation in these conditions (Brown et al., 2014)."

Line 476-477: This should not be presented as a conclusion.

We agree with your comments. We have deleted from the text.

Line 485-488: I think you undersell the importance of your work here, and you could make more compelling statements about the role of sea ice in CO2 fluxes in a changing Arctic.

We agree with your comments. We have deleted.

Figure 3. should this cite Hudson et al., 2015?

We agree with your comments. We have added "Hudson et al., 2015" in the Figure 3 caption.

Table 2. Consider adding an extra column for ΔpCO2 (air-sea difference) to aid under-standing.

Added as suggested.

Table 3. The key thing that jumps out for me is that natural flux is much higher for frost flowers than snow. I would have thought that's worth highlighting in your discussion.

We have added "and $F_{ff}$ was higher than $F_{snow}$, except for station FI1". We also indicated "Frost flowers are known to promote gas flux, such as $CO_2$, from the sea ice to the atmosphere (Geilfus et al., 2013; Barber et al., 2014; Fransson et al., 2015)."

Technical corrections
In general, the manuscript is well-written, the technical language is appropriate, and the standard of English is good. However, there are a couple of points to check throughout the text: use of the definite/indefinite article; singular/plural nouns and their following verbs e.g. frost flowers, was/were.

Thank you. During revisions we have tried to have our native-English co authors read through the text to improve the flow.

Line 84: should be transport by molecular diffusion

Changed accordingly.

Line 218: remove hyphens

Changed accordingly.

Line 377: I think 0.0 is a mistake.

$F_{snow}/F_{ice}$ ratio for FI6 was 0.02. Therefore, we indicated it as "0.0".

Table 3: brackets in the top line are confusing.

We have changed.

**Anonymous Referee #2**

The manuscript makes interesting observations of CO2 flux through sea ice, but re- quires extensive improvement. It was never articulated how this study is novel. I feel that it perhaps may be novel, but it is unclear how in its current form. Major revisions are needed before this manuscript can be considered publishable. The abstract is borderline uninformative. What are characteristic fluxes? Are these important? Of course CO2 can flux through sea ice, but it's hard for the reader to gage exactly how trivial this is without values in the abstract to justify reading the rest of the paper. The sentence beginning line 61 is a reference dump. What did these studies find and how does it build to the importance (or lack thereof) of the present manuscript?

We are grateful for your assessment of our work. We have now added some more discussion on the results from other work in the Arctic, and to emphasize the lack of observations in the pack ice:

We have added "In the ice covered Arctic Ocean, storm periods, with high wind speeds and open leads are important for air-to-sea $CO_2$ fluxes (Fransson et al., 2017), due to the under-saturation of the surface waters in $CO_2$ with respect to the atmosphere. On the other hand, the subsequent ice growth and frost flowers formation in these leads promote ice-to-air $CO_2$ fluxes in winter (e.g. Barber et al., 2014). Given the fact that Arctic sea ice is shrinking and shifting from multi-year ice to first-year ice, the area of open ocean and thinner seasonal ice is increasing. Therefore, the contribution of open ocean/thinner sea ice surface to the overall $CO_2$ fluxes of the Arctic Ocean is potentially increasing. However, due to the difficulty in acquiring observations over the winter pack ice, most of the winter $CO_2$ flux measurements were examined over the Arctic landfast ice. Therefore, there is a definite lack of information on conditions during wintertime, especially from Arctic pack ice." in introduction.

We have changed the final paragraph of the introduction "The Norwegian young sea ICE (N-ICE2015) campaign in winter and spring 2015 provided opportunities to examine $CO_2$ fluxes between sea ice and atmosphere in a variety of snow and ice conditions in pack ice north of Svalbard. Formation of leads and their rapid refreezing allowed us to examine air–sea ice $CO_2$ fluxes over thin young sea ice, occasionally covered with frost flowers in addition to the snow-covered older ice that covers most of the pack ice area. The objectives of this study were to understand the effects of i) thin sea ice and frost flowers formations on the air–sea ice $CO_2$ flux in leads, ii) effect of snow-cover on the air–sea ice $CO_2$ flux over thin, young ice in the Arctic Ocean during winter and spring seasons, and iii) of the effect of the temperature difference between sea ice and atmosphere (including snow cover) on the air–sea ice $CO_2$ flux." in introduction.

In the abstract, we have added $CO_2$ flux values "We found that young sea ice formed in leads, without snow cover, is the most effective in terms of $CO_2$ flux ($+1.0 \pm 0.6$ mmol C $m^{-2}$ $day^{-1}$) since the fluxes are an order of magnitude higher than for snow-covered older ice ($+0.2 \pm 0.2$ mmol C $m^{-2}$ $day^{-1}$)." We have added "Rare $CO_2$ flux measurements from Arctic pack ice show that two types of ice are significant contributors to the release of $CO_2$ from ice to the atmosphere during winter and spring: young thin ice with thin layer of snow, and old (several weeks) snow covered thick ice.".

68: Sea-ice CO_2 fluxes

Changed accordingly.

On line 81, please see Massman et al. (1995) as the fundamental reference on this topic (https://www.fs.fed.us/rm/pubs_exp_for/glees/exp_for_glees_1995_massman.pdf).

We agree with your comments. We have checked and added.

Somewhat harsh transition before the last paragraph of the introduction. Please state more clearly how the background materials presented tie directly to the proposed study and therefore what makes the present study novel. Material in section 4.3 could help. (note that there are also many reference dumps here. Please explain what the studies found; it is your job to make the reader's job easy (https://www.sesync.org/blog/the- writers-job).

We agree with your comment. Please see our response to your first comment "The manuscript makes interesting observations of $CO_2$ flux ……and how does it build to the importance (or lack thereof) of the present manuscript?".

on line 132, how was it ensured that placement of chambers did not perturb the pressure gradients in the snow? Creating pressure gradients can push CO2 out (or pull it in).

We agree with your comment. First, the chamber collar was inserted 5 cm into the snow and 1 cm into ice at frost flowers site to avoid air leaks between inside and outside of chamber. Then, chambers were installed over the collar. Therefore, placement of chamber on collar would avoid creation of pressure gradient. In addition, LI-COR 8100-104 chambers used in this study have carefully designed pressure vents to prevent pressure gradients and wind incursion from outside the chamber (Xu L., et al. 2006).

Xu L., et al. 2006. On maintaining pressure equilibrium between a soil $CO_2$ flux chamber and the ambient air. Journal of Geophysical Research. 111, D08S10, doi:10.1029/2005JD006435.

on 144, please see Bain et al. (2005) as a relevant reference for wind-induced effects (https://www.sciencedirect.com/science/article/pii/S0168192305001164)

Thank you. We have checked and added.

Frost flowers are first introduced in the paragraph beginning on line 146. One assumes that these are somehow important for CO2 flux? The notion was not previously introduced. (see line 360. This belongs in the intro). I agree with Reviewer 1 that the manuscript was prepared somewhat hastily.

We have added "In addition, Fransson et al. (2015) indicated that frost flowers promote $CO_2$ flux from the ice to the atmosphere." in the introduction. We also mentioned "$F_{ff}$" in the method section.

153: what is station FI6? Abbreviations are introduced before they are explained. It would help to explain the geography of the site before the measurements, also to ensure that measurements were made with a random design in mind.

We have changed "Air–sea ice $CO_2$ flux measurements were done over young ice (YI stations), first-year ice (FI stations), and old ice (multi-year ice) (OI station).". We also referred to the table where all the stations are listed.

Extensive English improvement is needed in section 2.3

We agree with your comment. The native English-speaking co-author has now edited section 2.3 and gone through the text.

On line 266, what does 'near-constant 0 C' mean?

We agree with your comment. We have changed to "near 0".

60.0 cm sounds rather specific for a measurement of snow which I assume has frequent small undulations, either at the snow surface or snow-ice interface in section 3.4, per day is not a SI unit, and diurnal patterns in the flux may make it difficult to scale from the native measurements (in the SI units of seconds) to the full day.

Snow is variable, but given that these are spot measurement we report to snow depth at site of measurement, as it is the local conditions that will affect the conditions at the measurement site ice surface. We would like to keep unit used in this study because sea ice $CO_2$ flux community used in the previous studies and it would be convenient for comparisons.

416: the abbreviation F was introduced far earlier.

Correct, (F) deleted from the sentence.

432: this is actually interesting. By focusing on the challenge of estimating gas transfer velocity, the manuscript has some novel features. These might be initial hypotheses for future work if causality can't be determined, but the mechanisms of sea ice/atmosphere gas exchange make for a more interesting analysis even if remaining questions are left.

We agree with your comments. We estimated gas transfer velocity for station FI6 and tank experiment. The gas transfer velocity for $F_{ice}$ at station FI6 is higher than that of tank experiment examined in Nomura et al. (2006) even with very similar$\Delta pCO_{2\,b-a}$ and brine volume fraction. Therefore, our results clearly indicated that temperature difference between sea ice surface and atmosphere would produce an unstable air density gradient and upward transport of air, thereby increasing gas transfer velocity. The comparison of the gas transfer velocity would be useful to evaluate the temperature effect on the air-sea ice $CO_2$ flux.

Figure 4: avoid simultaneous use of red and green in a figure.

We agree with your comments. We have changed.

[revised manuscript text omitted]

a. Data of first measurement after removal of snow or frost flower.

b. "−" indicates no data.

c. Number of measurements in bracket.

d. Data of station OI1 was not included.

---

## Referee Report (RR1)

The manuscript by Nomura et al has improved significantly since initial submission, and I am satisfied with their responses to the majority of the points that I made on the original version. There are a small number of minor remaining issues, which I detail below. Once these have been addressed, I believe that this work will be suitable for publication in biogeosciences.

I think you (the authors) miss an opportunity to emphasise the importance of this work and the broader implications of these findings in the context of the profound changes in Arctic climate and sea ice that are underway. The original version had more of this emphasis, and in my first-round review I suggested strengthening these points to get the most of this work. Contrary to being strengthened, these statements were removed in the latest version; I suggest reinstating them and giving them more emphasis in the discussion and conclusions. The abstract could also be strengthened significantly by including the importance and implications there as well.

The grammatical points that I raised in my first review have been addressed to some extent, but there are still many points throughout the text where the definite/indefinite article (a/the) is used incorrectly or not used when it should be. There are a small number of places where singular/plural nouns and their following verbs are still incorrect, e.g. frost flower formation (line 116, 132), was/were. I am not sure whether this is something that the journal addresses upon acceptance, so will leave that to the Editor.

Specific comments:

In the abstract, you define "old" as several weeks, whilst throughout the rest of the manuscript it refers to >1 year (e.g. line 150). Perhaps "older" would be better in the abstract (line 42)

Line 89: should be underlying soil?

Line 119-120: This sentence would be better phrased as something like "A potential consequence of this might be an increased contribution of open ocean surface and/or thinner sea ice to the overall CO2 fluxes of the Arctic Ocean". It is more nuanced than what you had. This section sets the scene nicely for you to emphasise the importance and implications of your work, as suggested above.

Figure 2: I think one of your arrows is in the wrong place – please check

Table 2: I fully understand and sympathise with the challenges of polar fieldwork! Your addition is fine, but I would suggest using "logistical constraints" rather than "technical reason".

Line 206: effect of snow and frost flowers?

Line 259-260: in the case of the VINDTA 3C, VINDTA stands for Versatile INstrument for the Determination of Total inorganic carbon and titration Alkalinity. Or in the case of the basic model, it stands for Versatile INstrument for the Determination of Titration Alkalinity.

Lines 262-267: What is the uncertainty on pCO2 values arising from uncertainty in DIC and TA?

Line 323-324: Cite table 2

Line 378-380: this sentence is unclear. Does >5% mean when brine volume fraction increases by 5%, or when brine volume fraction is greater than 5%? If the latter, permeability increases by an order of magnitude compared to what? Please clarify.

Line 394-397: this still implies that your statement is true for most or all stations, and *especially* for FI5, FI6 and YI1. It is only true for those stations, so that should be made clear. In fact, there are more stations for which there is no significant difference (or even where Fice is slightly lower, YI2) than where Fice is higher. I understand that FI5, FI6 and YI1 show the effect that you want to describe, so in this sentence you should just remove "especially" and focus on describing those stations.

Line 400-402: I suggest stating that this is consistent with your findings

Line 420: this number requires a significant figure, and would tell your story much better if presented as 0.02

Line 430: "as compared to that in the atmosphere" not required as ΔpCO2 incorporates atmosphere and ice.

Line 431-433: my comment at lines 262-267 is relevant here, and should be noted

Line 433: I suggest changing "in these conditions" to "at station FI3", for clarity.

Line 454: special should be spatial

Line 468: e.g. should be i.e.

Line 472: "magnitude for the" not required

Figure 6: in my opinion, you do not need to show temperature and temperature difference. It is temperature difference that is critical, so I would remove temperature itself to declutter the plot, as it doesn't add anything

Line 508: is $r^2 > 0.7$ from linear regressions? Is a linear fit most appropriate here, or would you get a much better correlation from a different model? In either case, the model used should be stated.

---

## Author Response (AR2)

**Point-by-point responses to Referee #1 and 2.**

Journal: BG
Title: CO$_2$ flux over young and snow-covered Arctic pack ice in winter and spring
Author (s): Daiki Nomura et al.
MS No.: bg-2017-521
MS Type: Research article
Date: 14 May 2018

We thank the reviewers for their valuable comments, which have helped us to improve the manuscript.

For clarity, the authors' responses are inserted as green text.

**Referee #1 (Sian Henley)**

The manuscript by Nomura et al has improved significantly since initial submission, and I am satisfied with their responses to the majority of the points that I made on the original version. There are a small number of minor remaining issues, which I detail below. Once these have been addressed, I believe that this work will be suitable for publication in biogeosciences. I think you (the authors) miss an opportunity to emphasise the importance of this work and the broader implications of these findings in the context of the profound changes in Arctic climate and sea ice that are underway. The original version had more of this emphasis, and in my first-round review I suggested strengthening these points to get the most of this work. Contrary to being strengthened, these statements were removed in the latest version; I suggest reinstating them and giving them more emphasis in the discussion and conclusions. The abstract could also be strengthened significantly by including the importance and implications there as well. The grammatical points that I raised in my first review have been addressed to some extent, but there are still many points throughout the text where the definite/indefinite article (a/the) is used incorrectly or not used when it should be. There are a small number of places where singular/plural nouns and their following verbs are still incorrect, e.g. frost flower formation (line 116, 132), was/were. I am not sure whether this is something that the journal addresses upon acceptance, so will leave that to the Editor.

We are grateful for your favorable assessment. We have made changes in response to all of your recommendations and edited the text improve the readability of the text.

Specific comments:

In the abstract, you define "old" as several weeks, whilst throughout the rest of the manuscript it refers to >1 year (e.g. line 150). Perhaps "older" would be better in the abstract (line 42)

Changed accordingly.

Line 89: should be underlying soil?

Changed accordingly.

Line 119-120: This sentence would be better phrased as something like "A potential consequence of this might be an increased contribution of open ocean surface and/or thinner sea ice to the overall $CO_2$ fluxes of the Arctic Ocean". It is more nuanced than what you had. This section sets the scene nicely for you to emphasise the importance and implications of your work, as suggested above.

Agree. We have added "Thus, a potential consequence may be increased contribution of open ocean surface and/or thinner sea ice to the overall $CO_2$ fluxes of the Arctic Ocean. The dynamics of the thinner ice pack, through formation of leads and new ice, will play an important role in the gas fluxes from the ice pack.".

Figure 2: I think one of your arrows is in the wrong place – please check

We have moved arrows for $CO_2$ chamber.

Table 2: I fully understand and sympathise with the challenges of polar fieldwork! Your addition is fine, but I would suggest using "logistical constraints" rather than "technical reason".

Changed accordingly.

Line 206: effect of snow and frost flowers?

Changed accordingly.

Line 259-260: in the case of the VINDTA 3C, VINDTA stands for Versatile INstrument for the Determination of Total inorganic carbon and titration Alkalinity. Or in the case of the basic model, it stands for Versatile INstrument for the Determination of Titration Alkalinity.

Changed accordingly.

Lines 262-267: What is the uncertainty on pCO2 values arising from uncertainty in DIC and TA?

We have added "The calculated $pCO_{2\,b}$ values varied within 1.7% when DIC and TA values were changed within the standard deviation ($\pm2$ $\mu mol\ kg^{-1}$)" in the text.

Line 323-324: Cite table

Changed accordingly.

Line 378-380: this sentence is unclear. Does >5% mean when brine volume fraction increases by 5%, or when brine volume fraction is greater than 5%? If the latter, permeability increases by an order of magnitude compared to what? Please clarify.

Agree. We have changed to "It has been shown that ice permeability increases by an order of magnitude when brine volume fraction is greater than 5% as compared to when the brine volume fraction is less than 5% (Golden et al., 1998; Pringle et al., 2009; Zhou et al., 2013). A brine volume fraction of 5% would correspond to a temperature of −5°C for a bulk ice salinity of 5 – the so called "law of fives" (Golden et al., 1998).".

Line 394-397: this still implies that your statement is true for most or all stations, and especially Äi0for FI5, FI6 and YI1. It is only true for those stations, so that should be made clear. In fact, there are more stations for which there is no significant difference (or even where Fice is slightly lower, YI2) than where Fice is higher. I understand that FI5, FI6 and YI1 show the effect that you want to describe, so in this sentence you should just remove "especially" and focus on describing those stations.

Changed accordingly.

Line 400-402: I suggest stating that this is consistent with your findings

Agree. We have added "This finding was consistent with the previous studies".

Line 420: this number requires a significant figure, and would tell your story much better if presented as 0.02

Changed accordingly.

Line 430: "as compared to that in the atmosphere" not required as ΔpCO2 incorporates atmosphere and ice.

Changed accordingly.

Line 431-433: my comment at lines 262-267 is relevant here, and should be noted

Changed accordingly.

Line 433: I suggest changing "in these conditions" to "at station FI3", for clarity.

Changed accordingly.

Line 454: special should be spatial

Changed accordingly.

Line 468: e.g. should be i.e.

Changed accordingly.

Line 472: "magnitude for the" not required

Changed accordingly.

Figure 6: in my opinion, you do not need to show temperature and temperature difference. It is temperature difference that is critical, so I would remove temperature itself to declutter the plot, as it doesn't add anything

Changed accordingly.

Line 508: is r2 > 0.7 from linear regressions? Is a linear fit most appropriate here, or would you get a much better correlation from a different model? In either case, the model used should be stated

Changed accordingly.

Other point:
The native English-speaking co-author has now edited through the text.

**Referee #2**

The revised manuscript represents a dramatic improvement but I feel that the manuscript could and should be improved further before the manuscript is publishable.

We are grateful for your favorable assessment. We have made changes in response to all of your recommendations and edited the text improve the readability of the text.

I still question the use of 'significant' on line 42. Of course it isn't a large flux globally, but it is significantly different from zero often times.

We agree. We have removed "significant" from the text.

Line 68/9 is still a reference dump. What did these different studies specifically find?

We agree. We indicated findings in the lower part of this paragraph.

On 137, please define what a frost flower is for readers unfamiliar with this term.

We have added "(vapor-deposited ice crystals that wick brine from the sea ice surface)".

Line 177 is increasingly well understood, but references would still help.

Changed accordingly.

263: this is still a bit suspicious; was a vent present on the chamber?

LI-COR 8100-104 chamber has pressure vents to prevent pressure gradients and wind incursion from outside the chamber (http://www.ecotek.com.cn/download/Manual-LI-8100A-V2-EN.pdf).

491: include units for salinity unless this is a ratio (and then please specify).

According to Unesco (1985), the practical salinity scale defined as conductivity ratio has no units. We would like to follow the definition by Unesco (1985).

600: effectively impedes. Please check the manuscript once more for English usage. (One more example, "the snow surface" on 762. More can be found.)

Agree. The native English-speaking co-author has now edited through the text.

The statement on line 639-642 would benefit from a table showing the representative fluxes from the different studies for reference for the reader.

We agree that new table showing flux value for each reference may help reader. However, the comparison and discussion for $CO_2$ flux measurement from each reference by different method and device are beyond the scope of this paper. We will try to do these comparison and discussion (method and device comparison etc) in the ECV-Ice: SCOR working group task (http://www.scor-int.org/SCOR_WGs_WG152.htm) in the future.

Shouldn't the equation on line 684 appear instead in the Methods section rather than the end of the Discussion?

We would like to keep this equation in the Discussion because we used this equation to discuss the concept for $
[revised manuscript text omitted]

Nomura Daiki 2018/5/9 14:58